# MEK inhibitors block growth of lung tumours with mutations in ataxia–telangiectasia mutated

Michal Smida[1,2,*], Ferran Fece de la Cruz[1,3,4,*], Claudia Kerzendorfer[1], Iris Z. Uras[1], Barbara Mair[1,3,4], Abdelghani Mazouzi[1], Tereza Suchankova[5], Tomasz Konopka[1,3,4], Amanda M. Katz[6], Keren Paz[6], Katalin Nagy-Bojarszky[1], Markus K. Muellner[1], Zsuzsanna Bago-Horvath[7,8], Eric B. Haura[9], Joanna I. Loizou[1] & Sebastian M.B. Nijman[1,3,4]

Lung cancer is the leading cause of cancer deaths, and effective treatments are urgently needed. Loss-of-function mutations in the DNA damage response kinase ATM are common in lung adenocarcinoma but directly targeting these with drugs remains challenging. Here we report that ATM loss-of-function is synthetic lethal with drugs inhibiting the central growth factor kinases MEK1/2, including the FDA-approved drug trametinib. Lung cancer cells resistant to MEK inhibition become highly sensitive upon loss of ATM both *in vitro* and *in vivo*. Mechanistically, ATM mediates crosstalk between the prosurvival MEK/ERK and AKT/mTOR pathways. ATM loss also enhances the sensitivity of KRAS- or BRAF-mutant lung cancer cells to MEK inhibition. Thus, ATM mutational status in lung cancer is a mechanistic biomarker for MEK inhibitor response, which may improve patient stratification and extend the applicability of these drugs beyond RAS and BRAF mutant tumours.

[1] Research Center for Molecular Medicine of the Austrian Academy of Sciences (CeMM), 1090 Vienna, Austria. [2] Central European Institute of Technology, Masaryk University, 62500 Brno, Czech Republic. [3] Nuffield Department of Clinical Medicine, Ludwig Institute for Cancer Research Ltd, University of Oxford, OX3 7FZ Oxford, UK. [4] Nuffield Department of Medicine, Target Discovery Institute, University of Oxford, OX3 7FZ Oxford, UK. [5] Academy of Sciences of the Czech Republic, Institute of Biophysics, 61200 Brno, Czech Republic. [6] Champions Oncology, Hackensack, New Jersey 07601, USA. [7] Institute of Pharmacology and Toxicology, University of Veterinary Medicine, 1210 Vienna, Austria. [8] Clinical Institute of Pathology, Medical University of Vienna, 1090 Vienna, Austria. [9] Department of Thoracic Oncology, H. Lee Moffitt Cancer Center and Research Institute, Tampa, Florida 33612, USA. * These authors contributed equally to this work. Correspondence and requests for materials should be addressed to S.M.B.N. (email: sebastian.nijman@ludwig.ox.ac.uk).

Lung cancer remains the leading cause of cancer-related deaths[1]. Sequencing studies have identified the major genetic drivers in the various subtypes of lung cancer, including adenocarcinoma, the most frequent lung cancer[2–5]. However, only a minority of lung tumours harbour mutations in activated kinases such as EGFR, ALK, ROS1 and RET that can be targeted with specific small molecule inhibitors. Some of these (for example, gefitinib, erlotinib, crizotinib) have shown moderate success in the clinic, extending survival by several months on average[6–8]. However, none of these drugs provides a curative therapy in advanced disease setting due to emerging secondary resistance.

The majority of lung cancer mutations, such as ATM (ataxia–telangiectasia mutated) loss-of-function mutations, which are found in 5–10% of patients, are not actionable with small-molecule inhibitors. Some of these so-called 'undruggable' mutations may result in vulnerabilities through synthetic lethal interactions that can be exploited with drugs[9].

The serine/threonine kinase ATM is well known for its involvement in the DNA damage response (DDR) and maintaining genome stability but it has also been implicated in other cancer-relevant processes including cell growth, metabolism and mitochondrial homeostasis[10]. The gene name is derived from the disease ataxia telangiectasia (A–T), a severe genetic disorder caused by homozygous germline mutations in the ATM gene. A–T patients are predisposed to cancer, particularly those of lymphoid origin, and ATM variants have also been associated with cancer predisposition[10,11]. More recently, cancer genome sequencing has revealed frequent ATM somatic mutations in a variety of solid tumours, including lung ($\sim$10%), pancreatic ($\sim$12%) and bladder ($\sim$4%) cancers[2,4,5,12,13]. Interestingly, mutual exclusivity with p53 mutations in lung cancer suggests that ATM loss-of-function can partially substitute for p53 loss[5].

MEK1/2 are kinases that regulate cell proliferation and survival. MEK inhibitors are currently in clinical development for a variety of cancers, including lung cancer, with mutations specifically in the oncogene RAS or its downstream signalling components (for example, BRAF), which occur in a large number of cancers. The first MEK inhibitor (trametinib) has recently been approved for treating BRAF mutant melanomas but in lung cancer results have not been as encouraging[14,15]. However, this does not rule out that MEK inhibitors (alone or in combination with other agents) are effective in a select lung cancer patient cohort. Thus, the identification of mutations that predict sensitivity to MEK inhibition, particularly a common one like ATM, remains of great interest and could have immediate clinical applications. MEK1/2 kinases have not been directly linked with DNA repair. Thus, our experiments reveal an unexpected link between growth factor signalling pathways and ATM loss, providing a strong rationale for testing MEK inhibitors in the context of ATM mutant tumours.

## Results

**A pharmacogenetic screen links ATM with MEK.** To search for synthetic lethal interactions between lung adenocarcinoma tumour suppressor genes and anti-cancer drugs, we employed an *in vitro* screening strategy using isogenic cell lines[16,17]. These isogenic cell lines only differ in the expression of one specific gene, thereby simplifying the interpretation of the screening hits. We employed the AALE lung bronchial epithelial cell line as a relatively benign lung cell type that has been immortalized with SV40 large T-antigen and hTERT, and can become tumourigenic upon KRAS or HRAS expression[18]. We compiled a list of 10 tumour suppressor genes (that is, APC, ATM, CDKN2A, ERBB4, NF1, PRKDC, PTEN, SMAD4, SMARCA4 and STK11) that recur

in human lung cancer based on publicly available tumour sequencing data and the Catalogue Of Somatic Mutations In Cancer (COSMIC) database[2,4,5,17]. As p53 and RB1 are inactive in AALE cells due to the expression of large T-antigen, these tumour suppressor genes were omitted. HRAS expressing cells were infected with validated shRNA vectors targeting each tumour suppressor gene and screened in a pooled format[16] against 106 diverse FDA-approved and experimental anti-cancer drugs (Fig. 1a, Supplementary Fig. 1, Supplementary Tables 1 and 2 and Supplementary Dataset 1).

The top synthetic lethal interaction hit identified in the screen involved knockdown of ATM, a DNA damage signalling kinase, and the Chk1 kinase inhibitor AZD7762 (ref. 19; Fig. 1a,b). This interaction is consistent with the known role of Chk1 as a critical downstream effector of the related DNA damage signalling kinase ATR, and suggests that ATM-deficient cells are more dependent on the ATR signalling axis.

The second strongest drug sensitivity hit in ATM knockdown cells was unexpectedly with a MEK1/2 inhibitor (PD0325901)[20] (Fig. 1a). Given the potential clinical value of such an interaction, we went on to investigate this further. To validate the hit, we first repeated the infection of AALE cells with the individual ATM shRNA in the absence of HRAS. The ATM shRNA alone did not inhibit proliferation but upon treatment with MEK inhibitor we observed almost complete inhibition of colony outgrowth, consistent with the screening result (Fig. 1c). Similar results were obtained in growth curve assays (Fig. 1d). Two functionally validated (Fig. 1e, Supplementary Fig. 2) independent hairpin sequences had the same effect, each showing a highly significant reduction in cell viability upon treatment with 1 μM PD0325901 compared with DMSO treatment (two-sided *t*-test, $P < 0.0001$, Fig. 1f). To ensure the synthetic lethal interaction with ATM knockdown was indeed due to inhibition of MEK, we used an alternative, FDA-approved and highly specific MEK inhibitor trametinib. Indeed, ATM knockdown cells were also markedly more sensitive to trametinib across a range of concentrations, indicating an on-target effect of the drugs (Figs 1g and 2a). Furthermore, overexpression of an inhibitor-resistant MEK1 mutant (MEK1-L115P)[21] in the ATM knockdown cells completely rescued the effect of trametinib (Fig. 2b), independently verifying an on target effect of the drug.

**ATM inactivation sensitizes lung cancer cell lines.** We next extended our analysis to four additional lung cancer cell lines (Fig. 2c,d). H460 cells carry a KRAS mutation but were unexpectedly highly resistant (IC50 > 10 μM) to trametinib and to TAK-733, another MEK inhibitor (Fig. 2a). However, upon ATM knockdown, H460 cells then became highly sensitive (IC50 < 100 nM). A similar sensitization was observed for the intermediate sensitive cell lines H322 (KRAS and BRAF wild type) and H1755 (BRAF mutant). This suggests that ATM knockdown can affect the response to MEK inhibition in both the presence and absence of KRAS or BRAF mutations, both of which are common in lung cancer. Importantly, sensitization by ATM knockdown was not observed in the naturally ATM mutant H1666 cell line, supporting an on-target effect of the used shRNA sequences (Fig. 2d). Furthermore, ATM shRNAs did not sensitize cells to a control, unrelated inhibitor of c-MET/ALK, crizotinib (Fig. 2d). Thus, ATM knockdown sensitized several lung cancer cell lines with different genetic backgrounds to MEK inhibition, and even augmented the response of already sensitive BRAF mutant H1755 cells by an order of magnitude.

As complementaryDNA add-back experiments are complicated by the large size of ATM (350 kDa) and poor transfectability of the lung cancer cell lines, we next employed RNA-guided

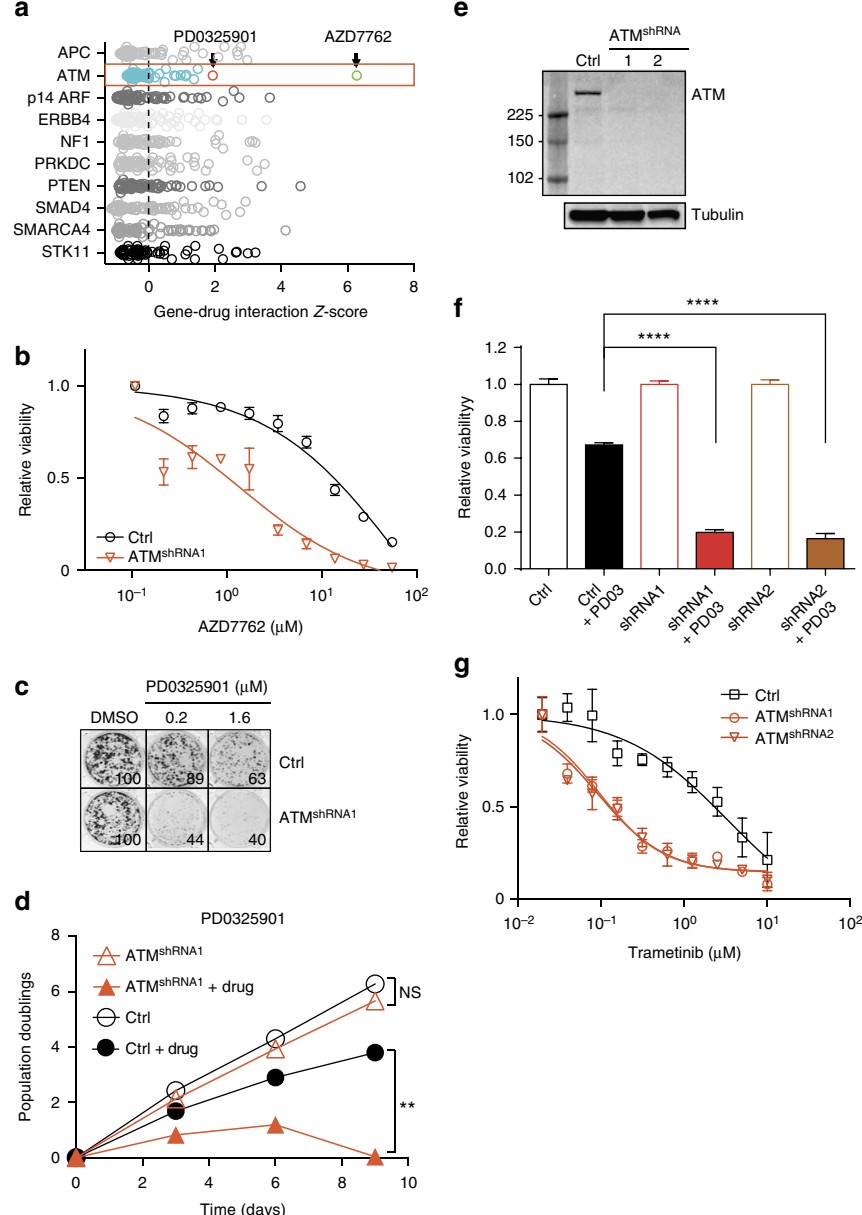

**Figure 1 | Chemical genetic screen reveals MEK inhibitor sensitivity of ATM-depleted cells.** (**a**) Gene–drug interaction screen in AALE cells (see Methods). Synthetic lethal drug interactions are sorted based on Z-score for each indicated tumour suppressor. The interactions between ATM and AZD7762 (green circle) and PD0325901 (red circle) are indicated with black arrows. (**b**) Dose–response curve with the Chk1/2 inhibitor AZD7762. AALE cells were infected as indicated and treated with the drug for 3 days. Displayed is the relative viability that is calculated by normalizing the raw CellTiter-Glo data to the vehicle (DMSO) treated controls. Error bars indicate s.d. ($n = 3$). (**c**) Colony formation of AALE cells infected with ATM shRNA or control virus and treated with PD0325901 for 10 days. Shown is a representative example ($n = 3$). Numbers in the bottom right corners indicate quantification relative to DMSO-treated samples. (**d**) Growth curves of indicated AALE cell lines treated with PD0325901 (1 μM). Cells were counted and passaged every 3 days and seeded at equal densities. $^{**}P < 0.01$, two-sided t-test ($n = 2$ biological replicates). (**e**) Western blot analysis of AALE cells infected with ATM knockdown vectors. (**f**) Cell viability of AALE cells infected with indicated vectors and treated with PD0325901 for 3 days. Data are normalized to vehicle (DMSO). Indicated are s.d.'s. $^{****}P < 0.0001$, two-sided t-test ($n = 3$). (**g**) Relative cell viability of AALE cells stably infected with ATM shRNA or control viruses and treated with trametinib for 3 days. Data are normalized to vehicle (DMSO). Error bars indicate s.d.'s ($n = 3$).

nucleases (RGNs)[22] as an alternative approach to inactivate *ATM* and confirm the synthetic lethal interaction. To avoid a bias due to *KRAS* or *BRAF* mutations, we selected H322 cells for this experiment. A CRISPR single guide RNA (sgRNA) targeting exon 6 of *ATM* was employed, and the genomic region flanking the Cas9 cleavage site in single cell clones was analysed by Sanger sequencing (Supplementary Table 3). *ATM* frameshift mutants were obtained at the expected ratios indicating that *ATM* was not required for viability. Furthermore, sequencing of

potential exonic sgRNA off target sites did not reveal any undesired gene editing (Supplementary Table 4). Complete absence of ATM protein expression was confirmed by western blot and non-edited wild-type clones obtained during the procedure were used as controls (Fig. 2e). As expected, these ATM-deficient NCI-H322 cells displayed a strongly reduced response to DNA damage as measured by phosphorylation of KAP1 and H2AX (Supplementary Fig. 3). In agreement with the knockdown experiments, *ATM* knockout NCI-H322 cells were markedly

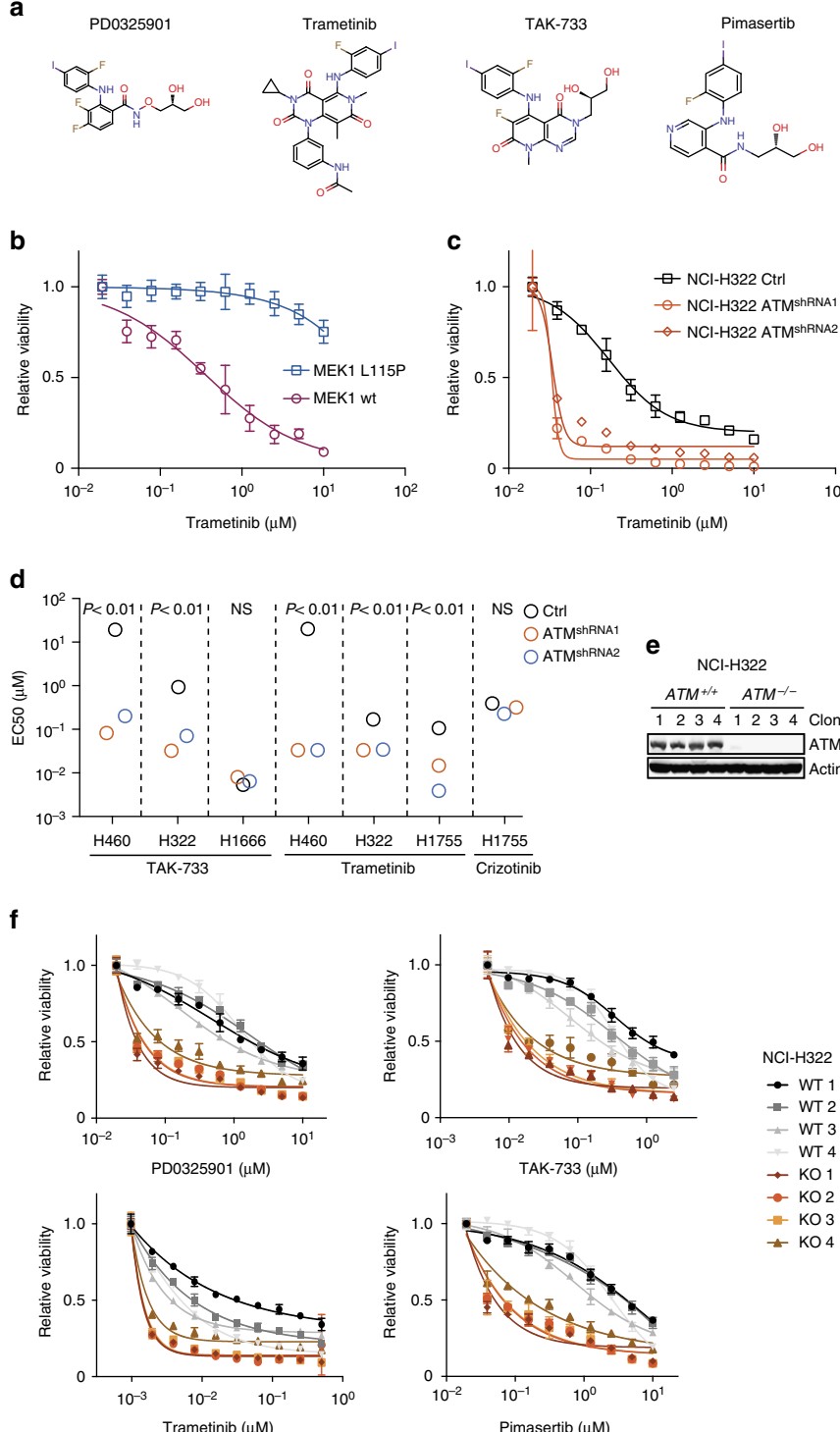

**Figure 2 | ATM loss-of-function in lung cancer cell lines triggers MEK inhibitor sensitivity.** (**a**) Chemical structures of MEK inhibitors used in this study. (**b**) Dose–response experiment of ATM knockdown (shRNA1) AALE cells infected with the indicated MEK1 expression vectors and treated with trametinib for 3 days. Data are normalized to vehicle-treated cells. Error bars indicate s.d.'s ($n = 3$). (**c**) Dose–response experiment of ATM knockdown NCI-H322 cells treated with trametinib for 5 days. Data are normalized to vehicle-treated cells. Error bars indicate s.d.'s ($n = 3$). (**d**) Effective concentration resulting in 50% growth inhibitory effect (EC50) is depicted for indicated cell line and compound combinations. The EC50 for the unresponsive control NCI-H460 cells was set to 20 μM. (**e**) Western blot analysis of CRISPR/Cas9 edited NCI-H322 clones. (**f**) Dose–response experiment of NCI-H322 cells in which both ATM alleles have been inactivated (four KO clones) or unedited control (four WT clones) treated with PD0325901, TAK-733, trametinib or pimasertib for 5 days. Data are normalized to vehicle-treated cells. Error bars indicate s.d.'s ($n = 3$).

more sensitive to MEK inhibition using four different inhibitors as compared with isogenic controls (Fig. 2f, Supplementary Fig. 4). Together these experiments demonstrate that ATM inactivation by shRNA knockdown or CRISPR/Cas9 knockout strongly sensitizes lung cancer cells to MEK inhibition even in the absence of *KRAS* or *BRAF* mutations.

**ATM mutations associate with sensitivity to MEK inhibition.** To investigate whether the ATM-MEK gene–drug association also occurs in lung cancer cells harbouring *ATM* mutations, we assembled a set of nine *ATM* mutant and seven wild-type control lung cancer cell lines. To control for confounding mutations, we included equal numbers of *KRAS/BRAF* mutant cell lines in the *ATM* mutant and wild-type groups and the status of the *ATM* mutant cell lines was confirmed by Sanger sequencing. As the *ATM* mutations have not been functionally validated we employed a polymorphism phenotyping algorithm (PolyPhen v2) to predict which mutations were most likely to affect protein function, for example by causing a premature stop codon or a missense mutation affecting an evolutionary conserved amino acid, and further stratify the cell lines[23]. In addition, we investigated the DDR by measuring the phosphorylation of the canonical ATM substrates SMC1 and KAP1 upon exposure to ionizing radiation (Supplementary Fig. 5). As expected, all *ATM* wild-type cells responded to ionizing radiation. In contrast, three out of the seven *ATM* mutant cell lines did not display an appreciable induction of SMC1 or KAP1 phosphorylation. This included the heterozygous *ATM* mutant cell line NCI-H1666. This experiment confirms that some of the cell lines with *ATM* mutations display aberrant DDR. However, it does not rule out that other aspects of ATM function are normal in the other cell lines that harbour *ATM* mutations[10].

Sensitivity to trametinib and TAK733 was determined in dose–response experiments and the area under curve (AUC) was calculated to compare the cell lines (Fig. 3a,b, Supplementary Fig. 6). There was a strong correlation (Spearman's correlation, rho = 0.94, $P < 0.0001$) between the potency of the two MEK inhibitors across the cell lines, indicating a shared mechanism of action (Fig. 3c). Out of the 16 cell lines, the top 4 most sensitive to both MEK inhibitors all carried *ATM* mutations. Furthermore, this enrichment of *ATM* mutations among cell lines with marked sensitivity was statistically significant even when only the biochemically validated cell lines were considered (two-sided *t*-test, $P < 0.01$, Supplementary Fig. 7A). In this panel, *KRAS* or *BRAF* mutational status did not significantly predict trametinib sensitivity. Yet, three out of the four most sensitive cell lines harboured *KRAS* or *BRAF* and *ATM* mutations, suggesting that cells with combined *ATM* and *BRAF/KRAS* are most likely to respond to MEK inhibitors. This is in agreement with the observation that MEK sensitive, *BRAF* mutant H1755 cells were further sensitized by ATM knockdown (Fig. 2d). Analysis of COSMIC (Catalogue of Somatic Mutations in Cancer) database indicates that *ATM* mutations and *BRAF* or *KRAS* (but not *TP53* and *EGFR*) mutations co-occur at a rate that is consistent with a lack of genetic interaction (Supplementary Fig. 7B).

To corroborate these results, we analysed data from the Cancer Cell Line Encyclopedia (CCLE)[24]. This public resource comprises a large panel of molecularly characterized cell lines, including 93 that are derived from lung cancer patients. The CCLE also contains information on the sensitivity of these cell lines to MEK inhibitors. The high frequency of *ATM* mutation in this collection (12 out of 93 cell lines, Supplementary Dataset 2) was similar to that previously reported in patients[2,4,5]. In agreement with our previous results, the mean response to the MEK inhibitors AZD6244 (selumetinib) and PD0325901 was significantly stronger in the *ATM* mutant cell lines (Fig. 3d, Supplementary Fig. 8). Remarkably, filtering out cell lines harbouring 'neutral' mutations (those predicted by PolyPhen not to affect protein function) preferentially removed MEK inhibitor-resistant cell lines and improved statistical significance 100-fold (Fig. 3d, Supplementary Fig. 8A). Similar results were obtained using a second independent data set from the Genomics of Drug Sensitivity in Cancer – COSMIC project[25] (Supplementary

Fig. 8B). This further strengthens the notion that it is specifically mutant *ATM* that contributes to MEK inhibitor sensitivity as filtering out random samples would have reduced statistical significance. Drug sensitivity was specific for *ATM* as no relationship between MEK inhibitors and an unrelated lung cancer tumour suppressor gene (that is, *ARID1A*) was observed (Fig. 3e). Furthermore, MEK inhibitors were the only 2 drugs out of the 20 tested that were significantly (two-sided *t*-test, $P < 0.01$) associated with *ATM* mutations (Fig. 3f), ruling out a general drug hypersensitivity phenotype due to ATM loss-of-function.

To further assess the performance of *ATM* as a predictive biomarker in the CCLE data set, we calculated sensitivity (fraction of responsive lines that is identified), false positive rates (fraction that is predicted to be responsive but is not) and true positive rates (fraction that is responsive and correctly predicted) and compared these with *KRAS/BRAF* (Supplementary Table 5). Encouragingly, *ATM* outperformed *KRAS/BRAF* in terms of true positive and false positive rate. Four out of seven (57%) of the cell lines predicted to be sensitive based on *ATM* mutation indeed responded to MEK inhibition. Importantly, a genetically stratified clinical trial with a 57% response rate would be considered successful (compared with the 23% that would have been observed using *KRAS/BRAF*). Adding *ATM* status to *KRAS/BRAF* further increased sensitivity and true positive rate, while reducing false positive rate.

Together our results reveal that a heterogeneous panel of lung cancer cells carrying *ATM* mutations is associated with high sensitivity to MEK inhibition.

To experimentally demonstrate that patient-derived mutations specifically in *ATM* are directly involved in MEK inhibitor sensitivity, we sought to restore resistance in *ATM* mutant cells. NCI-H23 cells have a homozygous *ATM* point mutation (c.5756A > C) resulting in an amino-acid substitution of glutamine with proline: ATM(Q1919P). Accordingly, these are highly sensitive to MEK inhibitors (Fig. 3a,b). We reasoned that if loss of ATM is causally implicated in MEK inhibitor sensitivity, restoration of the endogenous *ATM* locus would confer resistance to these compounds. To test this, we designed an sgRNA targeting *ATM* in close proximity of the c.5756A > C mutation. To enable restoration of the wild type allele, we delivered this sgRNA into NCI-H23 cells together with Cas9 and a DNA oligo encompassing the corrected wild-type sequence as a template for homology-directed DNA repair (Fig. 4a, Supplementary Table 1). Transfected cells were treated with MEK inhibitor so that any colonies that escaped drug treatment (for example, by restoring wild-type ATM) could be detected and analysed by sequencing. We observed restoration of mutant *ATM* to the wild-type allele in over 50% of the sequenced alleles, and only upon selection with trametinib or TAK733 (Fig. 4b). Thus, the *ATM* mutation found in NCI-H23 cells is required for the observed sensitivity to MEK inhibitors.

**Compromised crosstalk between MEK/ERK and AKT/mTOR.** Having established that ATM and MEK are synthetic lethal in lung cancer cells, we next investigated the mechanism underlying this gene–drug interaction. To determine whether ATM kinase activity is required for the MEK dependency, we used a specific small-molecule inhibitor of ATM (KU60019)[26] in combination with MEK inhibition. A single concentration of trametinib (40 nM) was chosen that had minimal effect on viability on its own. When combined with low ($< 1 \mu M$) concentrations of KU60019, we observed strong synergy as determined by Chou–Talalay combination index and deviation from Bliss additivity (Fig. 5a,b, Supplementary Fig. 9A). At higher concentrations

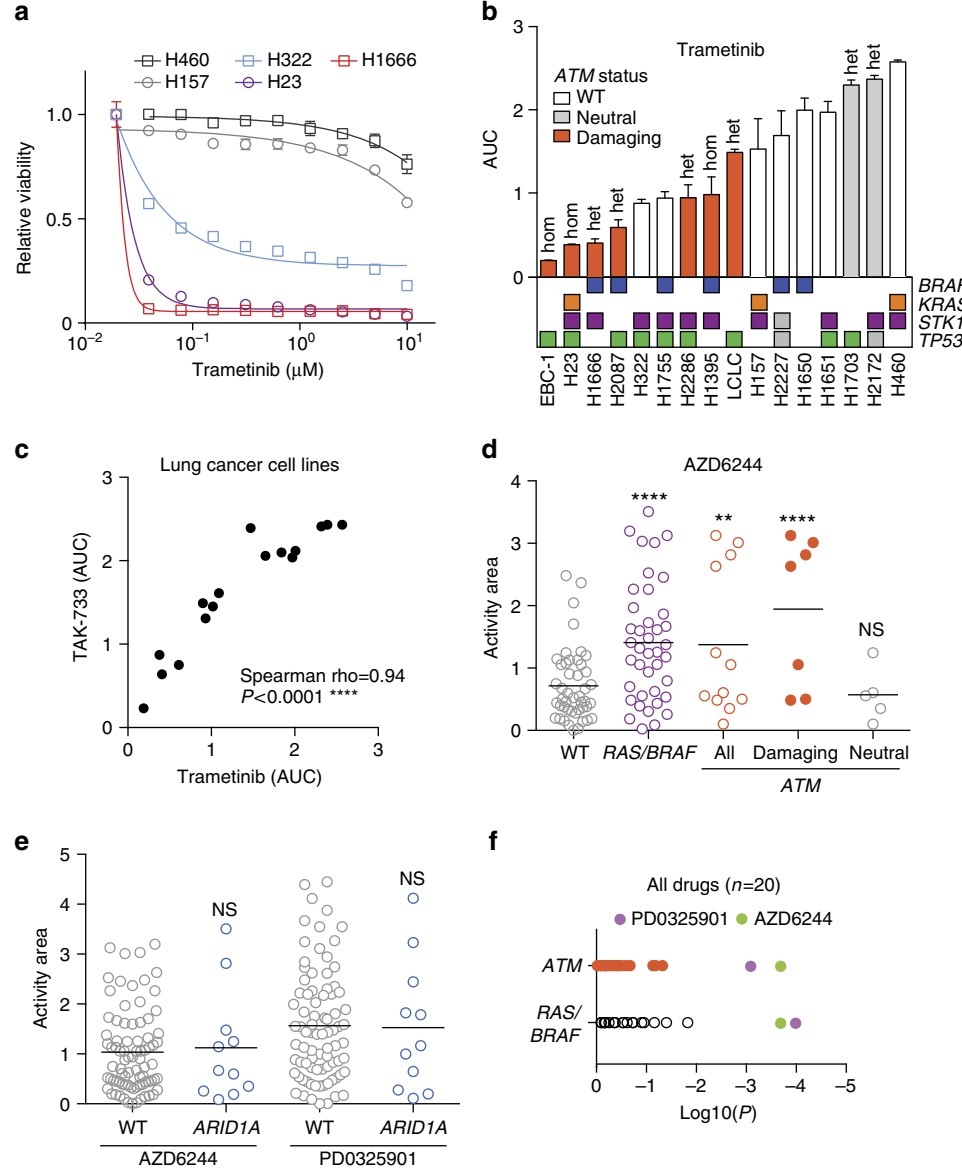

**Figure 3 | Cancer-associated ATM mutations predict MEK inhibitor sensitivity.** (**a**) Representative dose–response curves for sensitive and resistant lung cancer cell lines treated with trametinib for 5 days and normalized to vehicle control. Error bars indicate s.d.'s (*n* = 3). KRAS/BRAF genotypes for indicated cell lines: NCI-H460: KRAS-Q61H; NCI-H322: None; NCI-H23: KRAS-G12C; NCI-H1666: BRAF-G466V; NCI-H157: KRAS-G12R. (**b**) Sensitivity of indicated 16 cell lines to trametinib. Shown is the area under curve (AUC) derived from dose–response experiments as in **a**. When applicable, the heterozygous (het) or homozygous (hom) mutational status of ATM is indicated above the bars and mutational status for selected genes is indicated below. Error bars indicate s.d.'s (*n* = 3). (**c**) Area under curve values derived from dose–response experiments with TAK-733 and trametinib for 16 lung cancer cell lines. (**d**) Sensitivity of lung cancer cell lines in the Cancer Cell Line Encyclopedia (CCLE) to the MEK inhibitor AZD6244 (selumetinib). High activity area score (area above the curve[24] = AAS) indicates drug sensitivity. Each circle indicates a single cell line and cell lines are grouped according to genotype (WT = wild type for *K-Ras, H-Ras, N-Ras, BRAF, c-RAF* and *ATM; ATM = ATM* mutant; *RAS = K-Ras, H-Ras* or *N-Ras* mutant). *ATM* mutations are labelled according to PolyPhen predictions (damaging > 0.9, neutral < 0.9). Black bar indicates mean AAS. \**P* < 0.01, \*\*\*\**P* < 0.0001, NS = not significant, two-sided *t*-test compared with WT group. (**e**) Analysis as in **d** for *ARID1A* mutant or wild-type cell lines for sensitivity to indicated MEK inhibitors. NS = not significant, two-sided *t*-test. (**f**) Analysis of *ATM* or *RAS/BRAF* mutant cell lines for response to drugs (*n* = 20) in CCLE data set. Indicated is the *P* value for each drug.

(> 2 μM) KU60019 displayed some toxicity as a single agent (possibly due to off target effects) and synergy scores were accordingly lower. Thus, inhibition of ATM kinase activity renders lung cancer cells more sensitive to MEK inhibition.

We next analysed whether an altered DDR was involved in the synthetic lethal interaction. MEK inhibition in ATM null cells did not alter levels of the DNA double-strand-break marker γ-H2AX (Supplementary Fig. 10A). Moreover, MEK inhibition did not significantly alter phosphorylation of the ATM substrates

KAP1 or SMC1 upon induction of DNA double-strand breaks (Supplementary Fig. 10B). These results suggested an alternative explanation might be involved, such as a change in signalling through pro-survival or anti-apoptotic pathways. Indeed, upon exposure to MEK inhibitors, H322 ATM null cells underwent apoptosis, as evident from annexin V staining (Fig. 5c). This suggests that an increased propensity to undergo programmed cell death in response to MEK inhibition underpins the differential response between ATM wild-type and knockout cells.

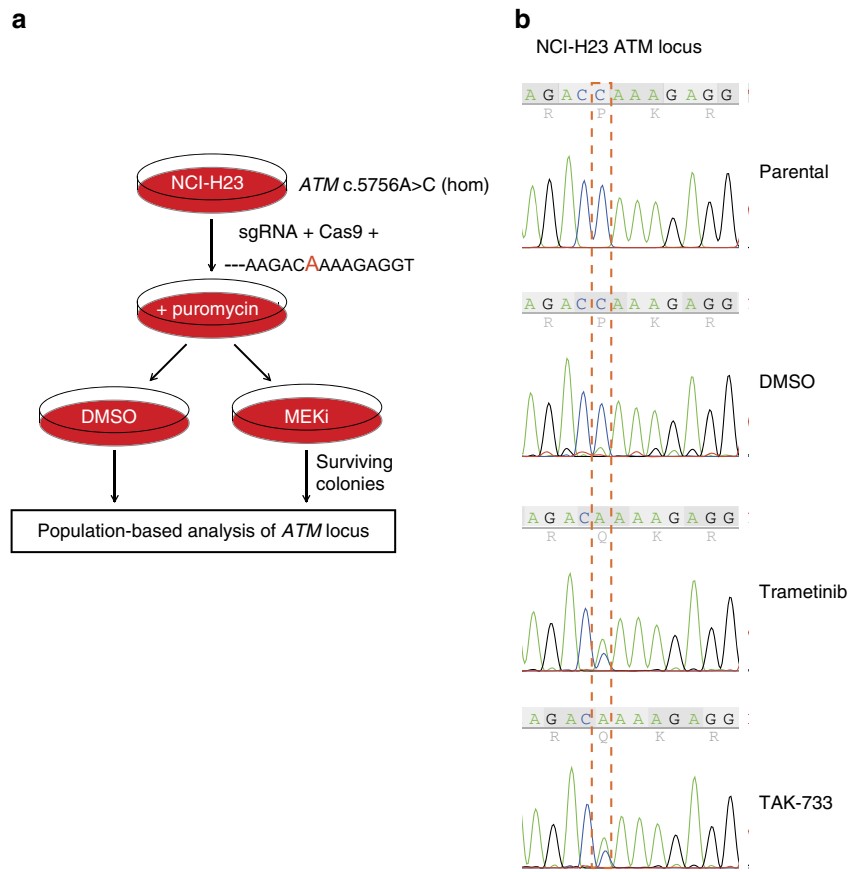

**Figure 4 | Restoration of ATM point mutation renders cells resistant to MEK inhibition.** (**a**) Schematic outline of CRISPR/Cas9 rescue experiment in *ATM* mutant NCI-H23 cells. (**b**) Sanger sequence chromatogram of cells treated as in **a**. Red dashed box indicates the mutant cytosine (C) that is replaced by adenosine (A).

One candidate pro-survival pathway is the PI3K/AKT/mTOR pathway, the activity of which is reciprocally linked to MEK/ERK signalling[27,28]. We treated isogenic NCI-H322 cells with MEK inhibitor for 6 h and determined the phosphorylation status of proteins along the AKT/mTOR pathway. In control cells, MEK inhibition resulted in increased phosphorylation of several components, consistent with abrogation of a negative feedback loop connecting the two pathways. Although pAKT itself was not elevated, phosphorylation of mTOR and 4EBP1 was increased in wild-type cells (Fig. 5d). Remarkably, this feedback mechanism was aberrant in *ATM* knockout cells. Instead of upregulation, we observed a downregulation of p4EBP1 and pS6K upon MEK inhibition whereas phospho-mTOR was unchanged and pAKT was slightly lower.

Next, we investigated if altered crosstalk was also observed in cell lines that naturally harbour *ATM* mutations. We subjected 13 wild-type ($n = 6$) and *ATM* mutant ($n = 7$) cell lines to MEK inhibitor treatment for 6 h and determined the phosphorylation status of 4EBP1, AKT, mTOR and S6K. The individual cell lines did not respond identically, as expected from a set of (epi-) genetically heterogeneous cell lines. Strikingly, we observed a significant difference between the two groups for 4EBP1, mTOR and S6K, where the ATM mutant cell lines consistently displayed diminished or inverted feedback response (Fig. 5f, Supplementary Fig. 11). A consistent reduction of pAKT in the absence or presence of MEK inhibition was not observed.

To determine whether reduced signalling through AKT/mTOR pathway would be sufficient to sensitize cells to MEK inhibitors, we measured potential synergy with the AKT inhibitor MK2206.

As before, concentrations of compounds were chosen that had minimal cytotoxic effects on their own. MEK and AKT inhibition strongly synergized in reducing cell viability (Fig. 5e, Supplementary Fig. 9B), in agreement with the previously reported observations in lung cancer models[29,30]. Importantly, this synergy was not observed in ATM-deficient cells, indicating that pro-survival compensatory signalling through the AKT–mTOR axis upon MAPK blockage requires functional ATM. In this context, signalling in ATM-deficient cells resembles the effect of an AKT inhibitor. Together, these results are consistent with a model in which ATM loss disturbs AKT/mTOR signalling thereby resulting in increased sensitivity to MEK inhibition.

**ATM loss-of-function sensitizes to MEK inhibition *in vivo*.** Next, we addressed whether the sensitization to MEK inhibition through ATM depletion in lung cancer cell lines was paralleled by a tumour response to MEK inhibitors *in vivo* by using mouse xenograft models. We first tested a patient-derived xenograft (PDX) model carrying a heterozygous *ATM* mutation (F858L) on the background of an activating KRAS mutation. This *ATM* mutation has been associated with increased radio-sensitivity and cancer susceptibility, indicating impaired signalling function[31,32]. Consistent with our previous results, the MEK inhibitor selumetinib completely blocked tumour growth and actually induced tumour regression (Fig. 6a). Selumetinib was more effective than vinorelbine and carboplatin/paclitaxel combination therapy, indicating a superior effect over several

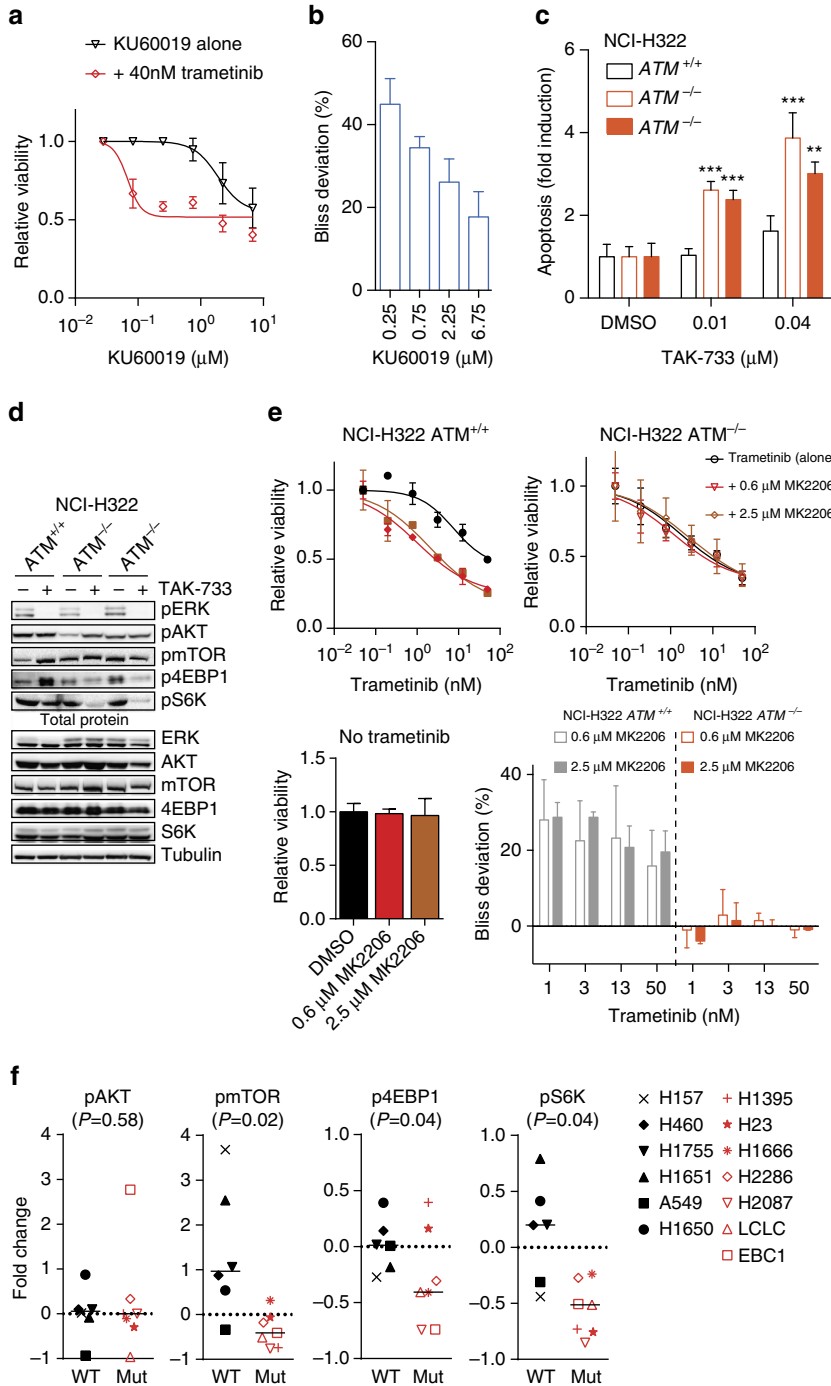

**Figure 5 | ATM loss-of-function dampens AKT/mTOR signalling. (a,b)** Drug synergy experiment in AALE cells with ATM inhibitor (KU60019) and trametinib (40 nM). Displayed is relative cell viability for KU60019 treatment alone and for co-treatment with trametinib. Error bars indicate s.d. ($n = 3$). **(b)** Deviation from Bliss additivity calculated from experiment in **a**. Error bars indicate s.d. ($n = 3$). **(c)** Fold induction of apoptotic cells after 3-day compound or vehicle (DMSO) treatment, as measured by annexin V positivity. Error bars indicate s.d.'s ($n = 4$). $^{**}P < 0.01$, $^{***}P < 0.001$, two-way ANOVA with Holm Sidak multiple comparisons correction. **(d)** Western blot analysis of *ATM* knockout and control NCI-H322 cells treated with TAK-733 (1.0 μM, 6 h) with indicated antibodies (pERK (T202/204); pAKT (S473); p-mTOR (S2448); p4EBP1 (T37/46); pS6K (T389)). Shown is a control and two independent knockout clones. For **d**, a representative blot for tubulin is used as loading control. **(e)** Drug synergy experiment on NCI-H322 control ( + / + ) and ATM knockout ( − / − ) cells treated with trametinib simultaneously with DMSO or two different concentrations of AKT inhibitor (MK2206). Relative cell viability is shown as dose–response curves. Effect of single MK2206 treatment on cell viability in the absence of trametinib is shown below as the bar graph. Deviation from Bliss additivity calculated for both control (grey bars) and knockout (red bars) cells for indicated concentrations is shown bottom right. Error bars indicate s.d. ($n = 3$). **(f)** Phosphorylation of indicated targets (pERK (T202/204); pAKT (S473); p-mTOR (S2448); p4EBP1 (T37/46); pS6K (T389)) in response to TAK-733 treatment (1.0 μM, 6 h) is shown as a fold change compared with DMSO-treated baseline (dashed line). Data were determined by quantification of digital western blot images in ImageJ for six wild-type (black symbols) and seven ATM mutant (red symbols) lung cancer cell lines. Median for each group is displayed as the horizontal line. Two-sided *t*-test was used to calculate the *P* values.

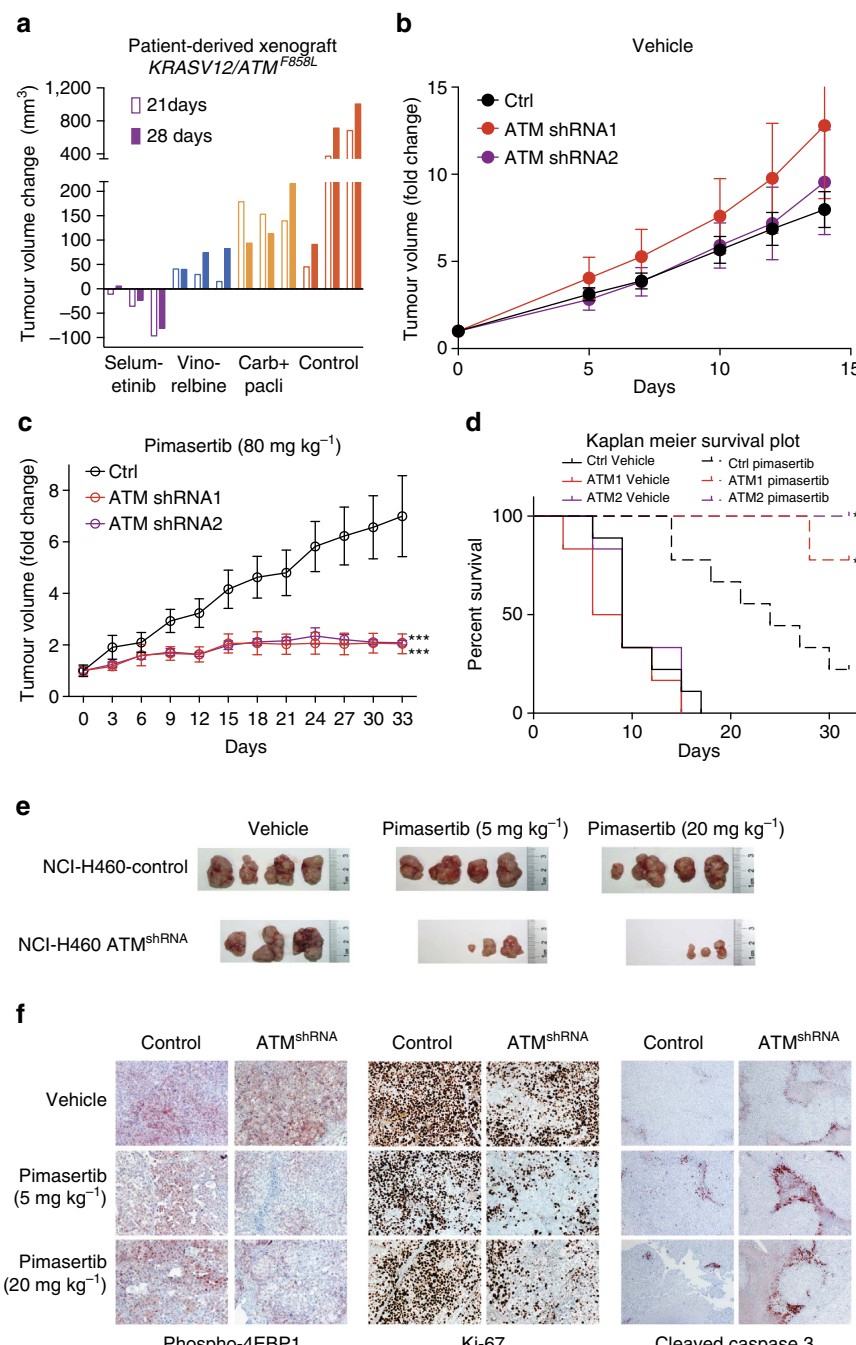

**Figure 6 | ATM depletion sensitizes tumours *in vivo* to MEK inhibition.** (**a**) Patient-derived xenograft model with *KRAS* and *ATM* mutations. Displayed is the change in tumour volume compared with baseline at 21 and 28 days post treatment for the indicated treatments. Day 21 control versus selumetinib $P < 0.05$; day 28 control versus selumetinib $P < 0.05$. Two-sided *t*-test ($n = 3$ biological replicates). (**b**) Growth of control, ATM shRNA1 and ATM shRNA2 NCI-H460 xenografts. (**c**) As in **b** treated daily with the MEK inhibitor pimasertib (80 mg kg$^{-1}$ p.o.). Shown is the mean and standard error. $^{***}P < 0.001$ ($n = 8$, two-sided *t*-test). (**d**) Kaplan–Meier survival curve of mice as in **b**,**c**. Control-pimasertib versus ATMshRNA1 or ATMshRNA2 $P < 0.05$, Log rank Mantel–Cox test. (**e**) Macroscopic images of excised H460 tumours treated as indicated daily for 14 days. (**f**) Immunohistochemistry staining of NCI-H460 tumours as in **e** at 14 days.

standard of care chemotherapeutics. Although this experiment supports a role of *ATM* in the sensitivity to selumetinib, lack of an isogenic control prevented us to rule out that other factors (for example, *KRAS*) are chiefly responsible for the response to MEK inhibition in this model.

To address this we next performed a xenograft experiment using an isogenic model. As the H322 cells did not graft in nude mice, we employed the MEK inhibitor-resistant H460 cells.

Furthermore, this allowed us to address the effect of ATM loss in the context of a KRAS mutation *in vivo* (Fig. 2d and Fig. 3a,b). Cells were transduced with control or one of two different ATM shRNAs and injected subcutaneously in nude mice. Functional validation of the hairpins confirmed an impaired DDR in the ATM knockdown cells (Supplementary Fig. 12). As expected, tumours grew at comparable speed indicating that the loss of ATM does not impact proliferation or survival on its own

(Fig. 6b). However, we observed strong sensitization of NCI-H460 cells to the MEK inhibitor pimasertib upon loss of ATM (Fig. 6c). Specifically, ATM wild-type tumours increased in size almost sevenfold over the course of the 33-day experiment. In contrast ATM null tumours markedly slowed down growth the first week upon commencing drug treatment and subsequently tumour volume remained stable until the end of the experiment. Importantly, pimasertib treatment also resulted in a significant survival benefit for mice carrying ATM knockdown tumours (Fig. 6d). Furthermore, the tumour inhibitory effects were already apparent at 5 mg kg$^{-1}$ pimasertib (Fig. 6e), which is more than 20-fold below the maximum tolerated dose of this compound.

The marked difference in response between control and ATM-depleted xenografts was also evident from immunohisto-chemistry analysis (Fig. 6f). Consistent with *in vitro* findings, we observed an increase in the numbers of apoptotic cells, as measured by cleaved Caspase 3 staining. Furthermore, upon MEK inhibition, tumours displayed a strong reduction in phosphory-lated 4EBP1 levels, consistent with a role for the AKT/mTOR pathway in the response to MEK inhibition. Expression of the proliferation marker Ki-67 was also strongly reduced in the ATM knockdown tumours upon MEK treatment, whereas only a minor reduction was observed in control tumours. This indicates that MEK inhibition in ATM-depleted tumours *in vivo* elicits a combination of proliferation arrest and apoptosis resulting in a strong inhibition of tumour growth.

## Discussion

The synthetic lethal interaction between ATM and MEK in lung cancer cells identified here indicates that these two kinases are functionally tightly linked. Our experiments suggest that this link does not relate to DNA damage signalling, the canonical function of ATM, but rather the known coordination between the MAPK and AKT/mTOR signalling pathways, which leads to an increased dependency on MEK kinase activity for cell survival. Thus, whereas most cells activate the AKT/mTOR pathway to compensate for loss of pro-survival signalling when the MAPK pathway is inhibited, ATM-deficient cells are unable to take advantage of this feedback loop. As a result, ATM-mutant lung cancer cells undergo apoptosis when MEK is inhibited. The notion that simultaneous inhibition of the MAPK pathway and the PI3K/AKT/mTOR axis is detrimental to cancer cells is supported by previous studies, although the mechanism has not been fully resolved[29,30]. However, this combination treatment results in a dose-limiting toxicity in patients[33].

Several other lines of evidence support the involvement of ATM in growth factor signalling, metabolism and the AKT/mTOR pathway in particular. For instance, *ATM* or *AKT* mutations in mice and humans both result in insulin resistance[34–36]. Furthermore, *ATM* mutant fibroblasts display reduced AKT/mTOR activity upon growth factor stimulation[37] and a recent report indicates that ATM supports oncogenic HER2 signalling in breast cancer cells[38]. A direct link between ATM and AKT/mTOR has also been suggested by the ability of ATM to phosphorylate 4EBP1 and PTEN[39,40]. In response to reactive oxygen species (ROS), reactive nitrogen species or temozolomide (an alkylating agent), ATM inhibits mTORC1 via LKB1, AMPK and ULK1, resulting in increased autophagy[41–45]. Interestingly, at least some of these effects take place in the cytoplasm, indicating a non-nuclear function of ATM in regulating metabolism. Furthermore, a similar role in metabolism was shown in the yeast *Aspergillus nidulans*[46], and ATM also localizes to mitochondria affecting mitophagy[47]. Together, these reports indicate an evolutionary conserved function of ATM in regulating metabolic homeostasis. Our experiments showing that ATM is required for crosstalk between the AKT/mTOR and MAPK pathways is in agreement with this role and provides new avenues to investigate ATM's function in growth factor signalling. Importantly, however, none of these studies have directly or indirectly hinted at hypersensitivity of ATM mutant cells to MEK inhibition.

Study of Atm heterozygous knockout mice have indicated that a 50% loss of Atm function does not result in a pronounced DNA repair phenotype[48]. This seems at odds with the observation that many of the ATM mutations in lung cancer are heterozygous and may thus only partially impair ATM. However, care must be taken to extrapolate these observations to somatic mutations in cancer, especially as most of these mutations have not been studied in any detail. Importantly, several observations show that absence of ATM protein does not equal presence of defective ATM. Indeed, ATM forms a homodimer and would thus be permissive to dominant negative effects of heterozygous mutations[49]. Similarly, mice with kinase dead Atm die before birth, whereas Atm null mice are viable[10,50,51]. Along the same lines, pharmacological inhibition of ATM does not phenocopy absence of ATM[52]. And human and mice carriers of ATM missense mutations (rather than truncation mutations that affect protein stability) have increased cancer incidence and can display dominant negative effects in cell line models[53–56].

The study of ATM function in cancer is further complicated by its involvement in other processes, including metabolic homeostasis as mentioned above. ATM is a large (350 kDa) protein and it is likely that different domains are critical for its various functions and it is conceivable that some ATM mutations impact these non-canonical functions while maintaining largely normal DNA damage signalling. Indeed, some of the MEK sensitive, ATM mutant cell lines responded to IR by phosphorylating KAP1 and SMC1. Although this may suggest normal ATM function in the DDR, other (not tested) functions of ATM could be affected in these cell lines. Thus, a seemingly normal response to IR in ATM mutant cells should not be interpreted as compelling evidence for normal ATM function. Inversely, however, absence of ATM kinase activity likely indicates a broad functional defect in ATM.

MEK inhibitors are currently being tested in clinical trials for efficacy in *RAS* or *BRAF* mutant lung cancer. However, these mutations alone do not adequately predict response to MEK inhibition[24], as also shown in this study. Furthermore, some *RAS* and *BRAF* wild-type lung cancer cell lines display strong dependency on MEK. Indeed, the most sensitive cell line in our panel (EBC-1) is *KRAS* and *BRAF* wild type but carries an *ATM* mutation. These observations indicate that the determinants of sensitivity to MEK inhibitors in lung cancer are still largely unresolved. Yet, unraveling the precise molecular requirements for MEK inhibitor efficacy will likely be a key determinant for the clinical success of these drugs in this highly challenging and genetically heterogeneous tumour type. Until now, only experimental compounds (for example, drugs inhibiting PARP[57–59], ATR[60]) have displayed a preferential toxicity in ATM loss-of-function cancer cells and none have been validated in lung tumours, where *ATM* is frequently mutated[2,4,5]. We show that *ATM* mutation in lung cancer cells results in a strong sensitization to drugs targeting MEK, including the FDA-approved drug trametinib. Thus, our findings suggest that including *ATM* mutational status in lung cancer as a mechanistic biomarker for MEK inhibitors can improve patient stratification, potentially extending the applicability of these drugs beyond *RAS* and *BRAF* mutant tumours.

## Methods

**Cell culture and general reagents.** AALE cells[18] were cultured in DMEM/F12 medium with 15% fetal bovine serum (FBS). All lung cancer cell lines were obtained from ATCC, except LCLC-103H cell line that was purchased from the DSMZ-German Collection of Microorganisms and Cell Cultures. All cell lines were maintained in RPMI medium with 10% FBS. All cells were grown in the presence of penicillin-streptomycin at 37 °C and 5% $CO_2$. The NCI-H157 cell line used in this study has been included in the database of commonly misidentified cell lines since it is suspected to be contaminated with NCI-H1264. We consider this fact irrelevant for the study conclusions since both cell lines are derived from lung carcinoma and both are KRAS mutant. Furthermore, this cell line was only employed to show the range of sensitivity to MEK inhibition in a diverse large set of lung cancer cell lines.

Phospho-SMC1 (S957) and gamma-H2AX (S319) antibodies were obtained from Millipore. SMAD4, ATM (2C1) and 53BP1 (H300) antibodies were purchased from Santa Cruz Biotechnology, β-actin from Sigma-Aldrich, phospho-KAP1 (S824) and KAP1 antibody from Bethyl Laboratories. All the other antibodies were from Cell Signaling Technology. All antibodies were used at a 1:1,000 dilution, except for α-tubulin that was used at a 1:5,000 dilution. Pemetrexed was obtained from Santa Cruz Biotechnology, TAK-733, trametinib, crizotinib, KU60019 and MK2206 from Selleck. Etoposide, paclitaxel, vinblastine, irinotecan, topotecan, gemcitabine, ifosfamide and neocarzinostatin (NCS) were purchased from Sigma. All other compounds were purchased from SynThesis Medchem (China).

MEK constructs were obtained from Addgene. To generate mutants, a QuikChange Site-Directed Mutagenesis Kit (Agilent Technologies) was employed. Introduced point mutations were verified by Sanger sequencing and mutants were shuttled into a gateway compatible pBABE-puro vector. All shRNAs were cloned into lentiviral pLKO.1-puro vector (Supplementary Table 1).

**Generation of isogenic cell lines and small-molecule screen.** shRNAs for tumour suppressors were introduced into cells by lentiviral transduction followed by puromycin selection. All of them were validated using western blotting or qRT–PCR. For the STK11 and NF1 isogenic cell lines, untransformed AALE cells were used. For all other tumour suppressors, HRAS-V12G transformed cells were employed. Stable cell lines were then individually tagged with DNA barcoded lentivirus and pooled[16].

For determining screening conditions, the drug concentration resulting in 50% AALE cell killing (IC50) was determined by performing 9-point dose–response experiments for all compounds. Based on these results, three concentrations were selected for the screen (IC50 and IC50 − / + 4-fold). One day after seeding pooled cells, drugs (or DMSO) were added in quadruplicates. After 6 days, genomic DNA was extracted and the barcodes were amplified by PCR using a biotinylated primer, labelled with streptavidin–phycoerythrin, and hybridized to Luminex xMAP beads coupled to the antisense barcode sequence. Samples were measured on a Flexmap 3D plate reader (Luminex). Data were analysed[16,61] using a linear regression-based method[16,61].

**Dose–response experiments and clonogenic and apoptosis assays.** Unless indicated otherwise, all cell viability assays were performed using the Cell Titer-Glo assay (Promega). Cells were counted and seeded in 96-well plates in triplicates. Next day, compounds were added and 3–5 days later, cell viability was assessed. AUC and EC50 determination was performed using GraphPad Prism. The percentage deviation from Bliss independency model was calculated by using the following formula:

$$E_{xy} = E_x + E_y - (E_x E_y)$$

Here, $E$ is the effect on viability of drugs x and y expressed as a percentage of the maximum effect.

Chou–Talalay drug combination (CI) indices were calculated by using the formula:

$$CI = (E_{drug1}/E_{ci}) + (E_{drug2}/E_{ci})$$

Here $E_{drug1}$ is the effect (in percent of maximum effect) of drug 1 alone, $E_{drug2}$ the effect of drug 2 alone and $E_{ci}$ is the effect of both drugs combined. A drug combination index <1 is considered synergistic.

For colony formation assay, 10,000 cells were seeded on six-well plates and treated with drug or vehicle control for ~3 weeks until clear colonies were formed. Colonies were fixed with 3.7% formaldehyde and stained with 0.1% crystal violet.

For determination of apoptosis, cells were treated with MEK inhibitor or vehicle and analysed for annexin V positivity (Biolegend) and DNA content (propidium iodide, Sigma) by FACS.

**Quantitative real-time PCR.** RNA was extracted using an RNeasy MinElute Cleanup kit (Qiagen). Isolated RNA was then subjected to DNAse treatment (Turbo-DNA free, Ambion). Reverse transcription was carried out using random hexamer primers and RevertAid reverse transcriptase (Fermentas). Quantitative real-time PCR was performed employing the KAPA SYBR FAST ABI Prism (Peqlab). Analysis was carried out in triplicates, using GAPDH as a control gene.

**Western blotting.** Cells were lysed in RIPA lysis buffer (50 mM Tris, 150 mM NaCl, 0.1% SDS, 0.5% sodium deoxycholate, 1% NP-40) supplemented with protease and phosphatase inhibitors. Lysates were sonicated, centrifuged and cooked with reducing sample buffer. Protein samples were separated by SDS–polyacrylamide gel electrophoresis on 4–12% gradient gels (Invitrogen) and transferred onto PVDF or nitrocellulose membranes. Quantification of band intensity on digital images was done in ImageJ and intensity of phosphorylation normalized to total protein staining. Uncropped images of all the blots in main figures are included as Supplementary Fig. 13.

**CRISPR/Cas9 mediated genome engineering.** Cells were plated at high density and co-transfected with a gBlock (IDT) encompassing the guide RNA (5′-GGATGCTGTTCTCAGACTGACGG-3′) expression cassette and a plasmid encoding the Cas9 nuclease[62]. Individual cell clones were expanded and the ATM target region located in exon 6 was amplified by PCR using flanking primers (fwd: 5′-GCGACCTGGCTCTTAAACTG-3′; rev: 5′-CAGAAAGTGTTGGACTTGG TTG-3′) and subsequently analysed by Sanger sequencing. Confirmation of monoallelic indels was determined by TA cloning of individual PCR products into a pCR 2.1 Vector (TA cloning KIT, Life Technologies) followed by Sanger sequencing of bacterial colonies.

To restore the mutated ATM allele in NCI-H23 cell line, cells were plated at high density and co-transfected with a plasmid encoding the Cas9 nuclease, a PCR product encompassing the guide RNA (5′-ACTACATGAGAAGACCA AAGAGG-3′) and a double-stranded 120 bp oligonucleotide containing the wild-type sequence (5′-TGCTGTTTGGATAAAAAATCACAAAGAACAA TGCTTGCTGTTGTGG ACTACATGAGAAGACAAAAGAGGTAATGTAAT GAGTGTTGCTTCTTACGTTTAGGATCTAGAG TGTAACTTGTT-3′). An oligonucleotide containing a silent codon change resulting in the same amino-acid substitution present in NCI-H23 was used as a control. After selection with puromycin, cells were plated and treated with DMSO, trametinib or TAK-733 for 3 weeks. DNA was isolated from surviving colonies and the mutated locus was amplified by PCR using flanking primers (fwd: 5′-CCCAGGCTAGTCAG TGAGTTC-3′; rev: 5′-GGAGCCAAGAAGGCTGCATAA-3′) followed by Sanger sequencing.

**Analysis of CRISPR off-target sites.** Potential off-target sites for the CRISPR sgRNA were predicted using the online tool crispr.mit.edu. Top five predicted exonic off-target sites were amplified from two independent NCI-H322 knockout clones and one control NCI-H322 wild-type clone by PCR using specific primers (TCP1- fwd: 5′-TGCGGGCA CAACATTATCCT-3′, rev: 5′-CTCAGTATT- CAGCCCTCAGCA-3′; GAPDH- fwd: 5′-TTCTAGGGTCTGGGGCAGAG-3′, rev: 5′-AAAACTATGCGAGGTGGGCA-3′; GALNT2- fwd: 5′-GAGAG GTGCCTGGCTTCTAC-3′, rev: 5′-GTGAAAGACAGAAGCGTGCG-3′; VAV1- fwd: 5′-CCAGCTCCTAGCAGTGTCTG-3′, rev: 5′-AGGAAGACGGG GACTCACAT-3′; CLEC9A- fwd: 5′-TGTTTTTGGGGGAGGTGATGT-3′, rev: 5′-TGTTGGCGTGTTAACCCTGA-3′). PCR products were cleaned up by ExoSAP (Affymetrix) and labeled with BrightDye Terminator Kit (Nimagen). Samples were purified by gel filtration through Sephadex resin (Sigma) and sequenced on ABI 3500 Genetic Analyzer (Applied Biosystems).

**CCLE and COSMIC data set analysis.** Data sets were downloaded from the respective data portal (that is, Broad Institute or Sanger/COSMIC) and mutation and drug sensitivity data was compiled in a single file. Only drug sensitivities were considered that had been tested on >80% of the lung cancer cell lines. All data was analysed in PRISM.

Polyphen analysis was performed using the online PolyPhen V2 tool. Mutations were considered damaging when the score was >0.9.

For mutation co-occurrence analysis, we looked at mutation profiles defined in the COSMIC database for cancer census genes (v70). We simulated randomized cohorts while keeping constant the empirical gene-wise mutation rates and the patient-wise mutation burden. Then, we compared the co-occurrence of mutations in ATM and other lung-cancer genes in the database and the simulated cohorts.

**Immunofluorescence microscopy.** Cells were plated onto coverslips (VWR) in a 24-well plate. Next day, cells were treated with 50 ng ml[−1] NCS or 2 μM TAK-733 for 30 min and 24 h, respectively, and DMSO treatment used as a control. Cells were allowed to recover for 2 h, washed twice with ice-cold PBS and fixed with 4% PFA + 0.1% Triton X-100 in PBS for 20 min on ice. Cells were permeabilized with 0.5% Triton X-100 in PBS for 20 min at room temperature and blocked with 10% FCS + 0.1% Triton X-100 in PBS for 1 h with always three washes between individual steps. Primary and secondary (Alexa Fluor 546 goat anti-rabbit and Alexa Fluor 488 goat anti-mouse; Invitrogen) antibodies were diluted in blocking solution and incubated for 1 h at room temperature. Finally, cells were stained with DAPI (1:1,000 in PBS, Sigma-Aldrich) for 20 min at room temperature in the dark. Cell images were acquired on a deconvolution microscope (Leica).

**Xenografts and immunohistochemistry.** Experimental procedures were approved by the Medical University of Vienna ethics committees and conform

to Austrian regulations. NCI-H460 (500,000 cells) were injected subcutaneously in nude mice and allowed to form palpable tumours before randomization and starting treatment with pimasertib. Number of animals (four for each treatment arm) in the study was chosen based on a large expected effect size. Animals that did not form tumours were excluded from the experiment. Drug was administered daily per oral gavage and tumours were measured using calipers and tumour volume was estimated using $V = 1/2(L \times W^2)$, where $L$ is the longest dimension (length) and $W$ is width (shortest dimension). The experiment was not blinded. For the Kaplan–Meier survival curves, animals bearing tumours larger than 1,000 mm$^3$ were considered as dead.

Ki67 immunohistochemistry stainings were prepared using a Ventana Benchmark Ultra automated staining device, applying the CONFIRM anti-Ki67 rabbit monoclonal primary antibody (clone 30-9, Ventana Medical Systems, Inc., Tucson, AZ) according to the manufacturer's protocol. Immunohistochemical stainings for cleaved Caspase 3 and phospho-4EBP1 (T37/46) (both from Cell Signaling Technology, dilution 1:200 and 1:1,000, respectively) were performed according to the manufacturer's protocol.

For PDX models, a tumour sample was removed from the tibia of a NSCLC patient and a tumourgraft model was generated. Selumetinib was dosed orally at 100 mg kg$^{-1}$, daily, for 28 days. Vinorelbine was dosed i.v. at 5 mg kg$^{-1}$ once a week, Carboplatin was dosed i.p. at 60 mg kg$^{-1}$ once a week and Paclitaxel was dosed i.v. at 10 mg kg$^{-1}$ once a week. All test agents were formulated according to the manufacturer's specifications. Beginning day 0, tumour dimensions were measured twice weekly by digital caliper and data, including individual and mean estimated tumour volumes (Mean TV ± s.e.m.), are recorded for each group. Tumour volume was calculated using the formula: TV = width$^2$ × length × $\pi$/6.

**Data availability.** The authors declare that the data supporting the findings of this study are available within the paper and its Supplementary Information files.

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

## Acknowledgements

We thank Richard van Jaarsveld, H. Lechtermann, Sejla Salic and J. Prochazkova for technical assistance; F. Ganglberger for help with data analysis; and F. Klepsch for help with graphics of chemical structures. We are grateful to W. Hahn for providing AALE cells and various plasmids; R. Weinberg for providing reagents; H. Pickersgill for editing and critical reading; and T. Brummelkamp for discussions. Pimasertib for *in vivo* studies was provided by Merck Serono, Merck KGaA, Darmstadt, Germany. The research leading to these results has received funding from the European Research Council under the European Union's Seventh Framework Programme (FP7/2007-2013)/ERC grant agreement no. (311166), the Moffitt Lung Cancer Center of Excellence and research grants from the Austrian Science Fund (FWF, P21768-B13) and the Vienna Science and Technology Fund (WWTF, LS09-009). T.S. was supported by a short-term fellowship (CZ.1.07/2.3.00/30.0030) and B.M. by a DOC fellowship of the Austrian Academy of Sciences. J.L. was supported by the European Union FP7 Career Integration Grant (PCIG11-GA-2012-321602) and an FWF Grant (P24766-B20) and M.S. was partly supported by the Ministry of Education, Youth and Sports of the Czech Republic under the project CEITEC 2020 (LQ1601).

## Author contributions

S.M.B.N. conceived the screen, supervised the project and analysed the data. M.S. and F.F.d.l.C. performed the majority of the experiments and analysed the data. C.K. performed most CRISPR/Cas9 experiments and performed dose–response experiments with shRNAs in lung cancer cell lines. I.Z.U. performed screen validation experiments and assisted in the DNA damage experiments. B.M. performed dose–response experiments. A.M.K. and K.P. performed the patient derived xenograft experiments. T.S. assisted with experiments involving the ATM inhibitors. K.N.-B. and Z.B.-H. performed and analysed immunohistochemistry experiments. M.K.M. assisted with screening set-up and data analysis. E.B.H. contributed to study design and data interpretation and J.I.L. assisted with DNA damage experiments. A.M. performed validation experiments of the ATM knockdown and knockout cells. With the help of the other authors S.M.B.N. wrote the manuscript. M.S., F.F.d.l.C. and S.M.B.N. assembled the figures.

## Additional information

**Competing financial interests:** The authors declare no competing financial interests.

