## [Peer Review File · Nature Communications]

Reviewers' comments:

Reviewer #1 (Remarks to the Author):

Based on a synthetic lethal screen involving a small selection of tumor suppressor genes with a group of FDA-approved anti-cancer drugs, the authors suggest that tumor cells with mutations in ATM exhibit increased sensitivity to Chk1 and MEK1/2 inhibition. The former result was not surprising and they then explore potential synergistic sensitivity of tumor cells containing ATM mutations or down-regulated ATM to MEK1/2 inhibition. While some potentially interesting relationships are noted in the manuscript, the degree of sensitization is relatively small in most experiments. Most problematic is that the data does not convincingly demonstrate that a given tumor cell could be predicted as ATM wild-type or mutant (heterozygous or homozygous - and there is a big difference between het and homozygous mutants) based on MEKi sensitivity; conversely, ATM status would not appear to convincingly predict which tumors would be sensitive to MEKi. Thus, it is not clear how this would be used clinically. It is also noted that the feedback loops between ATM status and the mTOR pathway is not new information (see C. Walker and colleagues publications) and that many of the experiments shown here have specific concerns that reduce confidence in biologically meaningful conclusions.

Specific Comments:

1. Figure 1 - 1C: No quantitation is shown for the clonogenic assays; 1E: they do not show the effects of PD alone or ATM shRNA alone on clonogenic survival; both alone would be expected to slow growth; "relative viability" is difficult to interpret quantitatively. Are the effects synergistic here? 1D: the western blot is insufficient to know that they are looking at ATM protein; it would be best to show full blots and the ATM signaling pathway.
2. Figure 2F - the differences are small and the contrived numbers are difficult to interpret. They would be better served by showing classic clonogenic survival curves, alone and in combination.
3. Figure 3 - Tumors with heterozygous ATM mutations, even if they are truncating or stop codon mutations, are probably physiologically indistinguishable from tumors that are wild-type for ATM (50% normal ATM function is virtually indistinguishable from 100% function in DNA damage response assays). ATM is a large gene and random mutations would be relatively common in tumors; as long as the mutation is heterozygous, ATM function would be relatively intact. The data in this figure is among the most important for arguing that ATM status would be an important predictor of MEKi sensitivity in patients - unfortunately, the lack of characterization of ATM functional status in the tumors and the spread of MEKi sensitivity across all genotypes precludes practical application of ATM mutational status as a determinant of using a MEK inhibitor. There are ATM-WT cells that are sensitive and ATM-mutant cells that are resistant in Fig. 3B. In Fig. 3C, there are 3 vs. 4 sensitive vs. resistant tumors with "damaging ATM" alleles; that may be statistically significant the way that they are calculating, but its biologic distinction is questionable. Thus, even if there is a statistically meaningful trend, the practical utility predicting which lung tumors would be sensitive to MEKi based on ATM status is not clear.
4. Figure 5 - Combination index and Bliss deviation may be elegant mathematical constructs, but they are highly derived calculations. They should show primary clonogenic survival data with each drug alone and in combination for easier biologic interpretation.
5. Figure 6 - 6A: The patient-derived xenograft is a heterozygous tumor; its ATM functional status is not characterized, making interpretation challenging. 6C: they need to demonstrate that their shRNA against ATM is able to reduce ATM levels and functionality for the full course of the experiment.

Other

1. They should show baseline and altered signaling pathways for MEK and ATM for every cell line tested.
2. Clinically relevant doses of trametinib would be less than 30 micromolar.
3. A potpourri of cell lines with different baseline sensitivities to MEK inhibition are used, but they are not characterized in these experiments and the variety make interpretation a bit more challenging.

Reviewer #2 (Remarks to the Author):

This is an interesting manuscript that finds that cancers with ATM mutations may have heightened sensitivity to MEK inhibitors. The authors utilize shRNA and CRISPR to demonstrate this in isogenic models. The finding has the potential to be impactful. The in vivo data and mechanistic data could be further developed.

Major Comments:

1) There are not a lot of mechanistic insights. And I do think that authors' interpretation of Figure 5D (in the Results and Discussion sections) is not entirely correct. In the ATM +/+ cells, there is actually no feedback on AKT phosphorylation demonstrated on the western blot shown. P-MTOR and P-4EBP1 may reflect upregulation of mTORC1 activity; but this may or may not be due to AKT activation. In fact, the more interesting finding may be that ATM -/- knockdown lowers basal levels of p-AKT and leads to direct regulation of mTORC1 by MEK signaling. But, to be clear, the effect of MEK inhibition of AKT phosphorylation (i.e., feedback) is not really different between the ATM +/+ and ATM -/- cells. To substantiate the authors' proposed mechanism, I think the authors should determine if PI3K/AKT signaling is generally less active in the lung cancer cell lines with endogenous ATM mutations and if they observe the same signaling in another isogenic pair. An actual mechanistic connection to this decrease in basal AKT signaling would be really interesting but could be the subject of another paper.

2) I think that the isogenic in vivo experiment is nice and necessary (I agree with the authors that a single PDX model fails to prove that in vivo sensitivity to MEK inhibitors is due to ATM loss). However, for the isogenic models, it is important to be sure that the MEK inhibitors are not synergizing with off-target effects of the shRNAs (as has plagued the field). Thus, the authors should rescue ATM expression in the shRNA lines by expressing a rescue cDNA that is resistant to knockdown. I think that this would make the data much more convincing.

Minor Comments:

1) For Figure 3A, I think the authors should test for ATM functionally as in Figure S2. It is unknown how accurate the PolyPhen v2 phenotyping algorithms truly are. What does a heterozygous mutation actually mean in terms of function of the DNA damage response?

2) For Figure 3A, what are the actual BRAF and KRAS mutations in each of the lines? For the BRAF mutations, there may be substantially different sensitivities to MEK inhibition based on the actual mutation.

3) Show EC50s for sensitivity to MEK inhibitors in isogenic H322 clones to complement Figure 2E. Also show raw data of viability curves in supplemental materials.

4) The use of H-RAS transformed cells for the screen should be explicitly referenced by the authors in the main text as it remains possible that the heightened sensitivity was due to the combination of H-RAS transformation and ATM loss.

5) Change "patient-derived cell lines" in the subsection heading. These are just regular human lung cancer cell lines that have been used for years and years in hundreds or thousands of papers. It is unclear why the authors would make them sound like something unique and derived from specific patients of interest.

6) Figure 3C-The figure legend should make clear that statistics are comparing each group to WT (I assume). How is activity area defined? What is the KRAS/BRAF status of the 4 ATM mutant cell lines that are sensitive? In this dataset, was ATM status an independent predictor of sensitivity taking KRAS and BRAF mutations into account?

Response to reviewers (original comments in *italic*). For clarity, we (re-)numbered the comments.

Reviewer #1 (Remarks to the Author):

1. *Based on a synthetic lethal screen involving a small selection of tumor suppressor genes with a group of FDA-approved anti-cancer drugs, the authors suggest that tumor cells with mutations in ATM exhibit increased sensitivity to Chk1 and MEK1/2 inhibition. The former result was not surprising and they then explore potential synergistic sensitivity of tumor cells containing ATM mutations or down-regulated ATM to MEK1/2 inhibition. While some potentially interesting relationships are noted in the manuscript, the degree of sensitization is relatively small in most experiments.*

We were actually very encouraged by the degree of sensitization that we observed. In most of the experiments with isogenic lines the sensitization is at least 5 –fold, and in many instances we note a 10 fold or greater sensitization (1G, 2C, D, F). In the field of oncology, this is regarded as strong sensitization (see for instance the recent paper by the group of Scott Lowe: Manchado et al. *Nature* June 2016) Importantly, the relevance of this degree of sensitization is demonstrated in the *in vivo* models that display substantial drug response.

2. *Most problematic is that the data does not convincingly demonstrate that a given tumor cell could be predicted as ATM wild-type or mutant (heterozygous or homozygous - and there is a big difference between het and homozygous mutants) based on MEKi sensitivity; conversely, ATM status would not appear to convincingly predict which tumors would be sensitive to MEKi. Thus, it is not clear how this would be used clinically.*

In non-cancerous cell lines (e.g., fibroblasts), genotype and drug sensitivity are often closely correlated (e.g., MMC hypersensitivity is a diagnostic hallmark for Fanconi Anemia). Such close correlations are historically rare in cancer cell lines that carry a plethora of mutations,

which also makes these types of experiments almost impossible to control. The degree of heterogeneity in cancer means that most studies that aim to correlate genotype to drug response use hundreds of cell lines to be able to arrive at statistically significant interactions.

This is a general and broadly observed phenomenon that has plagued the field for a long time. This is especially the case for solid tumours (e.g. lung cancer) and for drugs that do not directly target a mutated oncogene, as is the case here.

In clinical practice we want to be able to select those patients that are most likely to benefit from treatment with MEK inhibitors, while avoiding treatment of patients that are unlikely to respond. A reasonable assessment of the value of ATM as a biomarker for MEKi response would include comparing it to the current best clinical standard in the field i.e., KRAS/BRAF. To illustrate this more clearly and based on the CCLE cell line sensitivity data in Figure 3D, we have calculated sensitivity (fraction of responsive lines that is identified), false positive rate (fraction that is predicted to be responsive but is not) and true positive rate (fraction that is responsive and correctly predicted) for ATM, KRAS/BRAF and ATM+KRAS/BRAF.

	Sensitivity*	True positive	False positive
KRAS/BRAF	9/14 (64%)	9/39 (23%)	30/39 (76%)
ATM	4/14 (29%)	4/7 (57%)	3/7 (43%)
KRAS/BRAF/ATM	11/14 (78%)	11/41 (27%)	28/41 (68%)

*Activity area >2 is considered MEKi sensitive

Encouragingly, ATM clearly outperforms KRAS/BRAF in terms of true positive and false positive rate. Four out of seven (57%) of the cell lines predicted to be sensitive based on ATM mutation indeed responded to MEK inhibition. Importantly, a genetically stratified clinical trial with a 57% response rate would be considered successful (compared to the 23% that would have been observed using KRAS/BRAF). What our clinical collaborators are most interested in, moving forwards, is adding ATM status to KRAS/BRAF, which according to our data would further increase sensitivity and true positive rate, while reducing false positive rate. The numbers in the above table are not explicitly included in the current manuscript but we would be happy to do so.

3. *It is also noted that the feedback loops between ATM status and the mTOR pathway is not new information (see C. Walker and colleagues publications) and that many of the experiments shown here have specific concerns that reduce confidence in biologically meaningful conclusions.*

We are aware of the mentioned publications describing a connection between ATM and

mTOR signaling in response to reactive oxygen species and a link to autophagy, and now include additional references to this and related work in the discussion. The findings of the group of Cheryl Walker and others support the notion that ATM has functions outside of DNA damage signaling and impinges on growth factor signaling and survival pathways, as we also show in our manuscript. Thus, these papers support the notion of a role of ATM in regulating cell growth and apoptosis. However, we do believe our findings go significantly beyond the published work for the following reasons:

- Our data indicate a role of ATM in the crosstalk between the ERK/MEK and mTOR pathways. This has not been previously described.
- Our experiments are performed in the absence of agents that induce ROS. Almost all experiments in earlier papers concern effects in the presence of H₂O₂ or DNA damage.
- A link between ATM and mTOR as shown following ROS exposure does not imply hypersensitivity to MEK inhibition, also not in the absence of ROS, as we demonstrate here. Indeed, to our knowledge nobody has suggested that targeting the RAS/RAF/MEK pathway in ATM mutant tumors could be beneficial.

Specific Comments:

4. *Figure 1 - 1C: No quantitation is shown for the clonogenic assays;*

We now include a quantification of this experiment. In addition, to show that the shRNA alone has a very limited effect on proliferation we include a growth curve experiment with the control and knockdown cells in the absence of MEK inhibitor.

5. *1E: they do not show the effects of PD alone or ATM shRNA alone on clonogenic survival; both alone would be expected to slow growth;*

The effects of PD alone are shown in this figure (first versus second bar) and also in original Figure 1C (Ctrl sample). The effects of ATM shRNA alone are shown in the clonogenic assay in Figure 1C and in new Figure 1D (i.e., the growth curve). This experiment indicates no substantial difference in proliferation between the control and ATM shRNA transduced cells.

To indicate more clearly that the samples are normalized to the untreated control for each condition, we now show these samples (set to 1) in the updated figure (now 1F).

6. *"relative viability" is difficult to interpret quantitatively.*

Relative viability is determined by normalizing absolute viability (as obtained from CellTiterGlo) for the drug treated samples to the non-treated DMSO for each cell line. Thus, a 50% decrease in viability due to drug treatment would translate into a relative viability of 0.5, a 75% reduction in a relative viability of 0.25 and so on. We now explain this more

clearly in the legend corresponding to Figure 1B where it is first used.

7. *Are the effects synergistic here?*

Yes- the effects are synergistic as the ATM shRNA lines in the absence of drug are almost identical to the control. However, here we are using the term synergy for drug-drug combinations. Therefore, to avoid confusion we do not refer to this effect as synergistic.

8. *1D: the western blot is insufficient to know that they are looking at ATM protein; it would be best to show full blots and the ATM signaling pathway.*

We have followed the reviewer's suggestion and now include a larger blot with molecular weight markers. The shown band at the approximate correct molecular weight of ATM is strongly reduced upon transduction with either of the two ATM hairpins, which further supports it being the ATM protein. To test the functionality of the ATM knockdowns we now also include Western blots for downstream targets pSMC1 and pKAP1 upon addition of DNA damage (Figure S2).

9. *Figure 2F - the differences are small and the contrived numbers are difficult to interpret. They would be better served by showing classic clonogenic survival curves, alone and in combination.*

We agree that including original survival curves is useful and we therefore already had included these in the first submission as Figure S3. We have swapped the Figures and moved the AUC graph to the supplement.

The 2-fold change in area under curve (AUC) actually corresponds to a very substantial shift of the dose response curve. This can be appreciated when looking at the actual curves from which the AUC data was derived (Figure 2F). These were included in original supplementary figure S3.

AUC and Area Above Curve (AAC, also called "activity area") consistently capture differences in response whereas other metrics such as EC50 can be difficult to calculate when cells are completely resistant or extremely sensitive or when the shape of the curve is not sigmoid. Most large-scale studies investigating drug sensitivity of cell lines use AUC or AAC (see Garnett et al *Nature* 2012 and Barretina et al *Nature* 2012) to allow a standardized comparison between cell lines and between drugs.

10. *Figure 3 - Tumors with heterozygous ATM mutations, even if they are truncating or stop codon mutations, are probably physiologically indistinguishable from tumors that are wild-type for ATM (50% normal ATM function is virtually indistinguishable from 100% function in DNA damage response assays). ATM is a large gene and random mutations would be relatively common in tumors; as long as the mutation is heterozygous, ATM function would be relatively intact.*

This is certainly an interesting and relevant point to consider. It is, however, highly unlikely that ATM mutations in tumours are passengers as cancer re-sequencing studies correct for gene size and have found statistically significant recurrent heterozygous and homozygous mutations in ATM in a variety of solid tumours (Biankin *Nature* 2012; Imielinski et al. *Cell* 2012; CGAR *Nature* 2014a; CGAR *Nature* 2014b; Ding et al *Nature* 2008). Furthermore, these mutations are enriched in TP53 wild type tumours, further supporting functional relevance.

This may appear at odds with knockout mouse models and patient data that show that the DNA damage response is largely unaffected upon removal of a single ATM allele, reducing the amount of ATM in the cell by 50%. However, other published data supports the notion that heterozygous ATM missense mutations can result in defective tumour suppression:

- Mice carrying heterozygous missense *Atm* mutations (but not heterozygous *Atm* knockouts) display increased cancer incidence (Spring et al. *Nature Genetics* 2002).
- Human carriers of ATM missense mutations (rather than truncation mutations that affect protein stability) have increased cancer incidence (Stankovic et al *AJHG* 1999)
- Dominant negative effects of ATM missense mutations have been reported in cell line models (Scott et al *PNAS* 2002), in agreement with the notion that absence of ATM does not equal presence of kinase dead or chemically inhibited ATM (Yamamoto et al *JCB* 2012; Daniel et al *JCB* 2012; Shiloh *NRC* 2013; Choi and Bakkenist *Cell Cycle* 2010).

Newly added data in the manuscript also supports the notion that heterozygous ATM mutations can impinge on ATM function in the DNA damage response: The NCI-H1666 cell line displays a heterozygous ATM mutation but is largely defective in ATM signaling (as measured by pKAP1/pSMC1) and also hypersensitive to MEK inhibitors (Figure S5 and Figure 3).

Of course this does not rule out that some ATM mutations are passengers (as also suggested by our PolyPhen analysis that markedly improved statistical performance upon removal of non-damaging mutations). However, the fact that ATM is a bona fide tumour suppressor and the observation that missense mutations can affect ATM function even when heterozygous, makes it highly unlikely that the majority of mutations are passengers. We have also added a section concerning heterozygosity of ATM mutations in the discussion.

11. *The data in this figure [3] is among the most important for arguing that ATM status would*

be an important predictor of MEKi sensitivity in patients - unfortunately, the lack of characterization of ATM functional status in the tumors and the spread of MEKi sensitivity across all genotypes precludes practical application of ATM mutational status as a determinant of using a MEK inhibitor.

We agree that Figure 3 contains important data regarding the consistency of ATM mutational status as a predictor of MEK sensitivity, and have therefore worked to expand the data. First, we performed an additional control where we look at another lung cancer tumour suppressor gene (*ARID1A*) and show that there is no relationship between *ARID1A* mutational status and sensitivity to MEK inhibitors. This perhaps better highlights the value of the predictive capacity of *ATM*. We also look at an independent experimental dataset - the Sanger/COSMIC data - and find that, again, likely inactivating mutations in *ATM* make cells significantly more sensitive to MEK inhibitors. Finally, with the in-house cell line panel, we tested an additional MEK inhibitor (TAK733), and show that the sensitive cell lines were again enriched for validated functional *ATM* mutations. Of course, there are other known (*BRAF/KRAS*), and likely more unknown, mutations that are associated with sensitivity to MEK inhibitors. But for the first time, we are showing that MEK inhibitor sensitivity strongly and specifically associates with *ATM* mutations, in three independently generated data sets.

We agree that a full characterization of cancer-associated *ATM* mutations is an important goal. However, a full work-up of even a single mutation represents a formidable task especially given the technical difficulties of working with *ATM* plasmids and the likely great diversity of *ATM* functions (known and unknown) in addition to its role in DDR. As a first step, we do now include new data in the manuscript to address the functionality of *ATM* signaling in response to IR in the *ATM* mutant (and wild type) cell lines used in Figure 3. This experiment shows that the majority (4/7) of the *ATM* mutant cell lines (among them the heterozygous mutant NCI-H1666) have reduced pKAP/SMC1 upon exposure to IR (Figure S5). The remaining three appear to display normal *ATM* function in response to IR. However, this experiment does not address *ATM* functions other than the DNA damage response. We now include a brief discussion of this notion in the discussion section of the manuscript.

12. There are ATM-WT cells that are sensitive and ATM-mutant cells that are resistant in Fig. 3B.

We would not expect all *ATM-WT* cells to be resistant to MEK inhibitors because we know that at least *BRAF*, *KRAS* and *MEK* mutations also all drive dependency on MEK. So it is likely that the *ATM-WT* cells that are sensitive have a mutation in one of these, or a related pathway. Likewise, we would not expect all *ATM* mutations to be predictive of MEK sensitivity - because of the degree of heterogeneity in the cancer cells, some may be bystander mutations, and some may have their effect on MEK sensitivity blocked by other mutations that work downstream. The complexity of cancer cells means an all-or-nothing phenotype is highly improbable.

13. *In Fig. 3C, there are 3 vs. 4 sensitive vs. resistant tumors with "damaging ATM" alleles; that may be statistically significant the way that they are calculating, but its biologic distinction is questionable. Thus, even if there is a statistically meaningful trend, the practical utility predicting which lung tumors would be sensitive to MEKi based on ATM status is not clear.*

Again, the nature of the cancer cells means that an all-or-nothing phenotype is essentially unobtainable. Therefore, we are looking for a degree of significance that we can be confident is of biological/clinical relevance. Calculating that for individual experiments is difficult. But for example, as indicated in our response to point 2, ATM mutant status alone or in combination with KRAS/BRAF outperformed the current standard in the field. This fact, and taking into account all our additional data across the entire manuscript including the experiments in isogenic lines and the in vivo data gives us the confidence that this is strongly worth pursuing for clinical value, particularly given the next steps are relatively straightforward because MEK inhibitors have been FDA/EMA approved and ATM mutational status can be readily assessed in tumour samples.

14. *. Figure 5 - Combination index and Bliss deviation may be elegant mathematical constructs, but they are highly derived calculations. They should show primary clonogenic survival data with each drug alone and in combination for easier biologic interpretation.*

We felt that it was important to show that the combinations are synergistic in a mathematical sense. However, we appreciate that these numbers are not intuitive for all readers. Therefore, we now also include representative viability data.

15. *Figure 6 - 6A: The patient-derived xenograft is a heterozygous tumor; its ATM functional status is not characterized, making interpretation challenging.*

We agree that interpretation of this experiment is challenging as it lacks an isogenic control. We also point out this caveat in the text and perform additional in vivo experiments with a second model to address this.

As indicated in the manuscript, however, ATM-F858L has been shown to increase cancer susceptibility (Dork et al, Cancer Research 2001; Rudd et al, Blood 2006; Fletcher et al, CEBP 2010; Johnson et al. Human Molecular Genetics 2007; Meyer et al, Radiotherapy and Oncology, 2007) and correlate with increased radiosensitivity (Gutierrez-Enriquez et al. Genes, Chromosomes & Cancer 2004). This suggests that this variant at least partially impairs ATM function. Thus, even though we cannot make a definitive conclusion regarding the causality of this mutation and the sensitivity of the tumour to the MEK inhibitor selumetinib, we believe that the experiment supports the model and is therefore important to include.

16. 6C:they need to demonstrate that their shRNA against ATM is able to reduce ATM levels and functionality for the full course of the experiment.

For practical reasons we were not able to reanalyze the tumors in this experiment for ATM directly. However, we now include biochemical data to functionally validate the vectors in the cell line used in 6C (i.e., NCIH460, Figure S12).

Other

17. They should show baseline and altered signaling pathways for MEK and ATM for every cell line tested.

As suggested, we now include data on ATM pathways and pERK for all cell lines (Figure S5 and S11). We have also substantially extended our biochemical AKT/mTOR pathway analysis (Figure S11)

18. Clinically relevant doses of trametinib would be less than 30 micromolar.

We agree and throughout the manuscript we typically see responses in the nanomolar range.

19. A potpourri of cell lines with different baseline sensitivities to MEK inhibition are used, but they are not characterized in these experiments and the variety make interpretation a bit more challenging.

Using lung cancer cell lines to confirm the observations in isogenic models is a critical aspect of the validation. We agree that some of these lines have been better studied than others. We have tried to include relevant information regarding the used lines in the manuscript (e.g., mutations) and have investigated the lines for their response to IR. Furthermore, ATM mutations don't particularly cluster with the most common mutations observed in NSCLC. Therefore, variability in terms of mutations in our cell line panel we believe only strengthen the notion that ATM status is a determinant in the response to MEK inhibitors.

Reviewer #2 (Remarks to the Author):

This is an interesting manuscript that finds that cancers with ATM mutations may have heightened sensitivity to MEK inhibitors. The authors utilize shRNA and CSRPR to demonstrate this in isogenic models. The finding has the potential to be impactful. The in vivo data and mechanistic data could be further developed.

Major Comments:

20. *There are not a lot of mechanistic insights. And I do think that authors' interpretation of Figure 5D (in the Results and Discussion sections) is not entirely correct. In the ATM +/+ cells, there is actually no feedback on AKT phosphorylation demonstrated on the western blot shown.*

We agree that there is no increase in the levels of pAKT and apologize for the confusion. What was meant here is that in wild type cells the overall output activity of the “AKT/mTOR pathway” is increased upon MEK inhibition or at least remains stable. For instance, in most wild type cells lines we observe an increase in p-mTOR upon MEK treatment. We have re-phrased the section to clarify this.

21. *P-MTOR and P-4EBP1 may reflect upregulation of mTORC1 activity; but this may or may not be due to AKT activation. In fact, the more interesting finding may be that ATM -/- knockdown lowers basal levels of p-AKT and leads to direct regulation of mTORC1 by MEK signaling. But, to be clear, the effect of MEK inhibition of AKT phosphorylation (i.e., feedback) is not really different between the ATM +/+ and ATM -/- cells.*

We agree that the feedback is not at the level of pAKT itself, as hopefully clarified above and in the text. We do, however, not think that the primary mechanism is through lowering of pAKT as we did not observe a consistent pattern in the lung cancer cell line panel. Therefore, this may be a H322 specific effect rather than a broadly observed effect.

22. *To substantiate the authors' proposed mechanism, I think the authors should determine if PI3K/AKT signaling is generally less active in the lung cancer cell lines with endogenous ATM mutations and if they observe the same signaling in another isogenic pair. An actual mechanistic connection to this decrease in basal AKT signaling would be really interesting but could be the subject of another paper.*

Following the reviewer's suggestion, we now include also data from 13 additional wild type and ATM mutant cell lines. Although there is some variability across the panel we noted a significant reduction in the compensatory effects on the AKT/mTOR pathway in the ATM mutant lines (i.e, pmTOR, p4EBP1 and pS6K) upon MEK inhibition. This is now included in Figures 5F and S11.

23. *I think that the isogenic in vivo experiment is nice and necessary (I agree with the authors that a single PDX model fails to prove that in vivo sensitivity to MEK inhibitors is due to ATM loss). However, for the isogenic models, it is important to be sure that the MEK inhibitors are not synergizing with off-target effects of the shRNAs (as has plagued the field). Thus, the authors should rescue ATM expression in the shRNA lines by expressing a rescue cDNA that is resistant to knockdown. I think that this would make the data much more convincing.*

We certainly appreciate the suggestion to perform a rescue experiment. However, despite many attempts we were not successful at expressing ATM at sufficient levels in a reasonable fraction of the cells. This is due to the fact that the ATM cDNA is very large (>9kb, 3056 amino acids) and most of the lung cancer cell lines are difficult to transfect at a reasonable efficiency (>5%).

Overall, it appears highly unlikely that the effects are through an off target effect that would be the same for both shRNAs particularly as we also provide substantial data using CRISPR and chemical inhibition of ATM that all show the same effect.

We do, however, include more validation data of the H460 knockdown cell lines to confirm their functionality (Figure S12).

Minor Comments:

24. *For Figure 3A, I think the authors should test for ATM functionality as in Figure S2. It is unknown how accurate the PolyPhen v2 phenotyping algorithms truly are. What does a heterozygous mutation actually mean in terms of function of the DNA damage response?*

A full work-up of even a single mutation in ATM represents a formidable task especially given the technical difficulties of working with ATM plasmids and this likely great diversity of ATM functions in addition to its role in DDR. As a first step to address this point, we do now include new data in the manuscript to address the functionality of ATM signaling in response to IR in the ATM mutant (and wild type) cell lines used in Figure 3. This experiment shows that the majority of the ATM mutant cell lines have reduced pKAP/SMC1 upon exposure to IR (see Fig. S5). The others appear to display normal ATM function in response to IR but may be impaired in ATM functions not captured in this assay, under these specific conditions.

Of note, the heterozygous ATM mutant cell lines NCI-1666 displays impaired ATM signaling, indicating that mutation in a single allele can result in a near complete impairment of ATM

function. This is in keeping with several lines of evidence indicating that heterozygous missense ATM mutations can have altered function (also see reply Reviewer 1, point 10). Specifically, mice carrying heterozygous missense *Atm* mutations (but not heterozygous *Atm* knockouts) display increased cancer incidence (Spring et al. *Nature Genetics* 2002). Furthermore, dominant negative effects of ATM missense mutations have been reported in cell line models (Scott et al *PNAS* 2002), in agreement with the notion that absence of ATM does not equal presence of kinase dead or chemically inhibited ATM (Yamamoto et al *JCB* 2012; Daniel et al *JCB* 2012; Shiloh *NRC* 2013; Choi and Bakkenist *Cell Cycle* 2010).

25. *For Figure 3A, what are the actual BRAF and KRAS mutations in each of the lines? For the BRAF mutations, there may substantially different sensitivities to MEK inhibition based on the actual mutation.*

We now indicate the specific mutations (source CCLE) in the Figure legend: H460: KRAS-Q61H; H322: None; H23: KRAS-G12C; H1666: BRAF-G466V; H157: KRAS-G12R.

26. *Show EC50s for sensitivity to MEK inhibitors in isogenic H322 clones to complement Figure 2E. Also show raw data of viability curves in supplemental materials.*

We appreciate the suggestion and now include the raw viability curves for this figure.

27. *The use of H-RAS transformed cells for the screen should be explicitly referenced by the authors in the main text as it remains possible that the heightened sensitivity was due to the combination of H-RAS transformation and ATM loss.*

All validation experiments were performed in AALE cells without HRAS. We now include a comment concerning this in the text.

28. *Change "patient-derived cell lines" in the subsection heading. These are just regular human lung cancer cell lines that have been used for years and years in hundreds or thousands of papers. It is unclear why the authors would make them sound like something unique and derived from specific patients of interest.*

This was used to distinguish these lines from the AALE cells that are derived from normal lung tissue. However, we agree that this comes across as somewhat contrived and have adapted the text as suggested.

29. *Figure 3C-The figure legend should make clear that statistics are comparing each group*

to WT (I assume). How is activity area defined? What is the KRAS/BRAF status of the 4 ATM mutant cell lines that are sensitive? In this dataset, was ATM status an independent predictor of sensitivity taking KRAS and BRAF mutations into account?

We now mention in the Figure legend that the P value refers to a comparison to the WT group.

The definition of activity area (1-AUC) is derived from Barretina et al Nature 2012 and now more clearly indicated in the text of the figure legend.

There is some overlap in the KRAS/BRAF/ATM mutation status (Table S6). Two of the four most MEK sensitive, ATM mutant cell lines have a co-occurring KRAS mutation. Taking out the cell lines with co-occurring mutations results in a loss of power to come to a statistically significant result.

However, considering ATM in addition to KRAS/BRAF does result in an improvement in sensitivity (fraction of responsive lines that is correctly identified), false positive rate (fraction that is predicted to be responsive but is not) and true positive rate (fraction that is responsive and correctly predicted). Together, this shows that including ATM mutational status in cell lines improves the ability to predict MEK sensitivity.

Details are indicated in the below table.

	Sensitivity*	True positive	False positive
KRAS/BRAF	9/14 (64%)	9/39 (23%)	30/39 (76%)
ATM	4/14 (29%)	4/7 (57%)	3/7 (43%)
KRAS/BRAF/ATM	11/14 (78%)	11/41 (27%)	28/41 (68%)

*Activity area >2 is considered MEKi sensitive

The numbers in the above table are not explicitly included in the current manuscript but we would be happy to do so.

REVIEWERS' COMMENTS:

Reviewer #2 (Remarks to the Author):

The revised manuscript by Smida et al is substantially improved. The authors have addressed many of my initial comments. I have one remaining significant concern that I think the authors should address.

There appears to be some inconsistency between the data shown in Fig. S5 and Fig. 3b. In Fig. S5, several cell lines that have ATM mutations (predicted to be inactivating) have levels of pSMC1 and pKAP1 that are similar to those in cell lines with WT ATM. Importantly, among the 4 most MEK inhibitor sensitive cell lines in Figure 3B, two (EBC1 and H2087) have pSMC1 and pKAP1 levels that are similar to cells with WT ATM. Thus, these cells appear to have a functional ATM protein, and the heightened sensitivity to MEK inhibitors is therefore not related to ATM dysfunction. Perhaps, the authors could offer a clear explanation.

Reviewer #3 (Remarks to the Author):

Review Smida et al NCOMMS-16-01429A

This is an interesting manuscript describing the synthetic lethality of MEK inhibition and ATM inactivation (the latter through mutation, knockdown, or pharmacological inhibition) in lung tumor cells. The work might well identify a subgroup of lung cancers that respond favourably to MEK inhibitors.

Overall, this is a well-developed study, and the authors responded convincingly to the vast majority of reviewer comments.

Concerning Reviewer#1, I am inclined to disagree with the major points raised by the reviewer, and to agree with the responses of the authors. Specifically, I agree that the degree of sensitization towards MEK inhibitors seen by ATM inactivation is remarkable and encouraging. Moreover, I would not expect ATM mutation to be the one and only biomarker predicting sensitivity. Like the authors state, tumor cells are not as simple as that. Comparing isogenic cells with differential ATM status, as well as looking at the overall distribution of sensitivities across cell lines is the best one can do, and the authors have carried out all studies one could reasonably expect to support their point.

Having said this, I think it would be helpful to include the table sensitivity, true positive and false positive rates into the paper, not just in the rebuttal letter. It would, of course, be even more interesting to identify tumors that carry activating mutations in Ras or Raf, and simultaneously also an inactivating ATM mutation. Such "double" hits might predict the best response. But it is hard to predict how often such a constellation occurs. In the population reflected by the table, there should be five such patients, right? How did those respond to the MEK inhibitor? Are there any cell lines having both mutations? At least, the authors could discuss the feasibility of using the simultaneous presence of both mutations as a particularly stringent criterion for responding to MEK inhibitors.

In their response to Reviewer #1, point 10 (functional implication of heterozygous ATM mutations), it might be worth mentioning that ATM can form dimers to function properly (e. g. PMID: 12556884), helping to explain the dominance of an ATM mutant.

In my view, the weakest point about the manuscript is actually the mechanism that goes from MEK inhibition to AKT/mTOR/S6K activation, apparently requiring ATM. This point was also raised by Reviewer #2, but I still cannot see a clear answer. There have been reports saying that ATM phosphorylates PTEN and 4EBP1, as mentioned by the authors, but how is this induced by MEK inhibition? How does ATM mediate this without being activated towards its targets KAP1 and H2AX? Is PTEN even active in the cells under study (it is very often mutant in tumor cells)? Since I am not supposed to raise new points, I leave it to the discretion of Reviewer #2 and the authors to respond and discuss these questions.

Response to reviewers (original comments in *italic*). For clarity, we (re-)numbered the comments.

Reviewer #2 (Remarks to the Author):

1. *The revised manuscript by Smida et al is substantially improved. The authors have addressed many of my initial comments. I have one remaining significant concern that I think the authors should address. There appears to be some inconsistency between the data shown in Fig. S5 and Fig. 3b. In Fig. S5, several cell lines that have ATM mutations (predicted to be inactivating) have levels of pSMC1 and pKAP1 that are similar to those in cell lines with WT ATM. Importantly, among the 4 most MEK inhibitor sensitive cell lines in Figure 3B, two (EBC1 and H2087) have pSMC1 and pKAP1 levels that are similar to cells with WT ATM. Thus, these cells appear to have a functional ATM protein, and the heightened sensitivity to MEK inhibitors is therefore not related to ATM dysfunction. Perhaps, the authors could offer a clear explanation.*

Some of the ATM mutants display “normal” pKAP1 and pSMC1 under the conditions tested here. Importantly, however, this does not prove that ATM function is entirely normal. ATM is a large, multimodular protein and it is plausible that hypomorphic mutations can result in changes in specific ATM functions that are not tested in this particular experiment.

The lack of crosstalk between ERK/MEK and AKT/mTOR pathways in the EBC1 and H2087 cell lines (Fig S11) is consistent with the experiments in isogenic ATM mutant cells (Fig 5D) and therefore suggests a subtler ATM perturbation in EBC1 and H2087. This notwithstanding, it is not possible to completely rule out that the ATM mutations in these cell lines are passengers and that the perturbed cross talk is due to other mutations. We address this matter now more clearly in the discussion section on page 14 (lines 29-34).

Reviewer #3 (Remarks to the Author):

2. *This is an interesting manuscript describing the synthetic lethality of MEK inhibition and ATM inactivation (the latter through mutation, knockdown, or pharmacological inhibition) in lung tumor cells. The work might well identify a subgroup of lung cancers that respond favourably to MEK inhibitors. Overall, this is a well-developed study, and the authors responded convincingly to the vast majority of reviewer comments. Concerning Reviewer#1, I am inclined to disagree with the major points raised by the reviewer, and to agree with the responses of the authors. Specifically, I agree that the degree of sensitization towards MEK inhibitors seen by ATM inactivation is remarkable and encouraging. Moreover, I would not expect ATM mutation to be the one and only biomarker predicting sensitivity. Like the authors state, tumor cells are not as simple as that. Comparing isogenic cells with differential ATM status, as well as looking at the overall distribution of sensitivities across cell lines is the best one can do, and the authors have carried out all studies one could reasonably expect to support their point.*

We thank the reviewer for these supportive remarks.

3. *Having said this, I think it would be helpful to include the table sensitivity, true positive and false positive rates into the paper, not just in the rebuttal letter.*

As suggested, we now include the true/false positive analysis in the manuscript (Supplementary Table 7).

4. *It would, of course, be even more interesting to identify tumors that carry activating mutations in Ras or Raf, and simultaneously also an inactivating ATM mutation. Such “double” hits might predict the best response. But it is hard to predict how often such a constellation occurs. In the population reflected by the table, there should be five such patients, right? How did those respond to the MEK inhibitor? Are there any cell lines having both mutations? At least, the authors could discuss the feasibility of using the simultaneous presence of both mutations as a particularly stringent criterion for responding to MEK inhibitors.*

We would indeed predict that “double” mutant cell lines (and patients) would be most sensitive. Four characterized cell lines display ATM and KRAS/BRAF mutations (H2087, H1666, H23 and H1373). Notably, three of these cell lines were tested by us and are in the top four of most MEK sensitive cell lines (Figure 3B, H1373 was not available to us and tested as resistant in the CCLE). Despite this trend, this experiment is not sufficiently powered to arrive at statistical significance.

Regarding feasibility of using both mutations as a stringent criterion, we performed an analysis of the COSMIC patient data to investigate the frequency of “double” mutations. This analysis suggests absence of a genetic interaction between ATM and KRAS/BRAF. In other words, these mutations co-occur at a frequency that one would predict based on their individual frequencies (new Supplementary Figure 7B and text on page 7, lines 17-25). Thus, already in the UK alone some 500-2000 patients (i.e, 1-4%) would display compound ATM and KRAS/BRAF mutations.

5. *In their response to Reviewer #1, point 10 (functional implication of heterozygous ATM mutations), it might be worth mentioning that ATM can form dimers to function properly (e. g. PMID: 12556884), helping to explain the dominance of an ATM mutant.*

We now include reference to this.

6. *In my view, the weakest point about the manuscript is actually the mechanism that goes from MEK inhibition to AKT/mTOR/S6K activation, apparently requiring ATM. This point was also raised by Reviewer #2, but I still cannot see a clear answer. There have been*

reports saying that ATM phosphorylates PTEN and 4EBP1, as mentioned by the authors, but how is this induced by MEK inhibition? How does ATM mediate this without being activated towards its targets KAP1 and H2AX? Is PTEN even active in the cells under study (it is very often mutant in tumor cells)? Since I am not supposed to raise new points, I leave it to the discretion of Reviewer #2 and the authors to respond and discuss these questions.

We appreciate that many questions regarding mechanism remain unanswered. In light of the points raised by Reviewer 2 and the potential clinical relevance of our findings we are of the opinion that the precise mechanistic elucidation is beyond the scope of this manuscript.